# ScreenParse: Moving Beyond Sparse Grounding with Complete Screen Parsing Supervision

A. Said Gurbuz[1,2]   Sunghwan Hong[*,2,3,4]   Ahmed Nassar[1]   Marc Pollefeys[2,5]   Peter Staar[1]

## Abstract

Modern computer-use agents (CUA) must perceive a screen as a structured state, what elements are visible, where they are, and what text they contain, before they can reliably ground instructions and act. Yet, most available grounding datasets provide sparse supervision, with *insufficient* and *low-diversity* labels that annotate only a small subset of task-relevant elements per screen, which limits both coverage and generalization; moreover, practical deployment requires efficiency to enable low-latency, on-device use. We introduce **ScreenParse**, a large-scale dataset for *complete* screen parsing, with dense annotations of all visible UI elements (boxes, 55-class types, and text) across 771K web screenshots (21M elements). ScreenParse is generated by **Webshot**, an automated, scalable pipeline that renders diverse urls, extracts annotations and applies VLM-based relabeling and quality filtering. Using ScreenParse, we train **ScreenVLM**, a compact, 316M-parameter vision language model (VLM) that decodes a compact ScreenTag markup representation with a structure-aware loss that upweights structure-critical tokens. ScreenVLM substantially outperforms much larger foundation VLMs on dense parsing (e.g., 0.592 vs. 0.294 PageIoU on ScreenParse) and shows strong transfer to public benchmarks. Moreover, finetuning foundation VLMs on ScreenParse consistently improves their grounding performance, suggesting that dense screen supervision provides transferable structural priors for UI understanding. Project page: https://saidgurbuz.github.io/screenparse/.

[1]IBM Research Zurich, Zurich, Switzerland [2]ETH Zurich, Computer Vision and Geometry (CVG), Switzerland [3]ETH AI Center, Switzerland [4]ETH Zurich, Photogrammetry and Remote Sensing (PRS), Switzerland [5]Microsoft, Switzerland. Correspondence to: Sunghwan Hong[*] <sunghwan.hong@ai.ethz.ch>.

*Proceedings of the 43rd International Conference on Machine Learning*, Seoul, South Korea. PMLR 306, 2026. Copyright 2026 by the author(s).

## 1. Introduction

The rise of vision language models has opened a new era of computer use agents capable of interacting with graphical user interfaces (GUI) to perform complex tasks (Wang et al., 2025a; Qin et al., 2025; He et al., 2024; Zhang et al., 2025). Despite rapid progress, a fundamental bottleneck persists: the *grounding* problem (Cheng et al., 2024; Feizi et al., 2025). To operate effectively, a screen agent must first accurately identify UI elements, understand their roles, and reason about their spatial and functional relationships. This structural understanding is a prerequisite for effective downstream planning and action execution; when it fails, errors cascade throughout the agent pipeline.

Current state-of-the-art models for GUI interaction predominantly rely on "sparse" action-oriented datasets that annotate only the single element relevant to each task step (Deng et al., 2023; Zhou et al., 2024; Rawles et al., 2023; Xie et al., 2024). Such supervision is valuable for end-to-end policies, but it leaves the majority of on-screen elements unlabeled and the full screen structure implicit. As a result, models can learn shortcuts that are sufficient for the supervised steps while failing to form a complete screen state, which can hurt robustness and generalization to new layouts, applications, and out-of-distribution screens. In addition, practical deployments often require low-latency, on-device inference, motivating compact perception models rather than relying exclusively on large foundation VLMs (Bai et al., 2025; Zhu et al., 2025).

We argue that a natural remedy is to treat **complete screen parsing** as a core training objective. We define screen parsing as recovering the complete semantic structure of a screen: the set of all visible UI elements, their bounding boxes, semantic types, and associated text. Compared to single-target grounding, dense screen parsing provides a holistic screen understanding that downstream agents can condition on for instruction following and action selection.

A key challenge is that dense, complete annotations are expensive to obtain by human annotators and difficult to maintain at scale, especially on the web where pages are dynamic and Document Object Model (DOM)-derived elements can be noisy, redundant, or visually irrelevant. To

address this, we introduce **Webshot**, a scalable pipeline that renders diverse web pages and extracts dense DOM-aligned UI annotations, then applies VLM-based refinement and quality filtering to produce a high-quality dataset. Fig. 3 summarizes the Webshot pipeline.

Leveraging our Webshot pipeline, we construct **ScreenParse**, a large-scale dataset for complete screen parsing that provides dense annotations for all visible UI elements, including their bounding boxes, semantic types, and text, spanning 55 UI categories. Building on this data, we train **ScreenVLM**, an ultra-lightweight vision–language model that parses full screens into a structured sequence representation, ScreenTag. To better align optimization with structured extraction, we further introduce a structure-aware weighted loss that upweights structure-critical tokens *e.g.,* tags and locations, improving the fidelity of predicted layouts.

Empirically, ScreenVLM substantially outperforms much larger foundation VLM baselines on dense parsing and transfers effectively to public benchmarks. Moreover, we demonstrate that ScreenParse supervision benefits other model families as well, strengthening both foundational VLMs and detector-based parsers. These results suggest that dense screen supervision provides transferable structural priors for robust UI understanding.

In summary, our key contributions are:

- We introduce **ScreenParse**, a large-scale dataset for *complete* screen parsing, providing dense annotations of all visible UI elements, such as bounding boxes, element types, and text, across 55 UI categories.

- We propose **Webshot**, a scalable and fully automated pipeline that collects dense, hierarchy-preserving screen parsing annotations from rendered web pages.

- We show that training on **ScreenParse** yields strong and transferable gains: our proposed **ScreenVLM** architecture, as well as existing foundation VLMs and state-of-the-art detector-based parsers, improve substantially on both our benchmark and public UI understanding benchmarks.

**Conflict of Interest Disclosure.** Some authors are affiliated with IBM Research and Microsoft. The paper evaluates public baselines including OmniParser, a Microsoft-developed GUI parsing system, and introduces ScreenParse and ScreenVLM as part of the authors' research. The authors declare no financial conflicts of interest beyond these affiliations.

## 2. Related Work

**Computer-Use Agents and Evaluation.** Recent benchmarks evaluate end-to-end agents that perceive screens and execute actions in web and OS environments, spanning interactive tasks and demonstration-based settings (Zhou et al., 2024; Koh et al., 2024; He et al., 2024; Xie et al., 2024; Deng et al., 2023). More recent suites (Wang et al., 2025b) further expand evaluation to structured, multi-platform protocols. Although critical for assessing long-horizon success, these benchmarks often leave a fine-grained perception implicit; our work targets this gap by enabling dense screen-level supervision for both training and evaluation.

**UI Grounding Benchmarks and Datasets.** A closely related line of work studies UI grounding, where models localize elements referred to by natural-language instructions. SeeClick and ScreenSpot/ScreenSpotPro popularize instruction-conditioned grounding evaluation, and a concurrent work, GroundCUA, provides more complete screen-level annotations derived from human demonstrations (Cheng et al., 2024; Li et al., 2025; Feizi et al., 2025). However, most benchmarks offer sparse supervision (one instruction to one or a few elements), while more complete datasets like GroundCUA are limited in scale and diversity. In contrast, our dataset targets *complete* screen parsing with dense annotations of nearly all visible UI elements after rendering, providing a holistic perception prior that complements instruction-level grounding.

**Foundation VLMs and Parsers.** Foundation VLMs such as Qwen3-VL, InternVL3, and Gemini-2.5 can be prompted for grounding and structured extraction and are common baselines for GUI perception (Bai et al., 2025; Zhu et al., 2025; Comanici et al., 2025). Several UI-specialized VLMs also target GUI grounding and screen understanding, including CogAgent (Hong et al., 2023) and the Ferret-UI family (You et al., 2024; Yang et al., 2025). In parallel, specialized parsers such as OmniParser localize UI elements via detector-based pipelines (Lu et al., 2024). In practice, foundation VLMs are often too large for low-latency or on-device deployment, while detector-style parsers focus on localization and are commonly paired with a language model for downstream action selection. Our work is complementary: ScreenParse provides dense full-screen supervision that trains an ultra-compact VLM for complete screen parsing and also improves foundation VLMs and detector-based parsers on our benchmark and public evaluations (Feizi et al., 2025; Cheng et al., 2024).

## 3. Dataset: ScreenParse

### 3.1. Overview

This section introduces *ScreenParse*, a web-scale dataset for *complete screen parsing*, the task of recovering all rendered visible UI elements on a screen together with their locations, semantic types, and text. Unlike most GUI agent and grounding datasets that provide sparse supervision for only

the interacted or instruction-referred element(s) (Deng et al., 2023; Zhou et al., 2024; Rawles et al., 2023; Cheng et al., 2024; Xie et al., 2024), ScreenParse provides dense, screen-level annotations that encourage holistic screen understanding and make it possible to train parsers that generalize across diverse layouts.

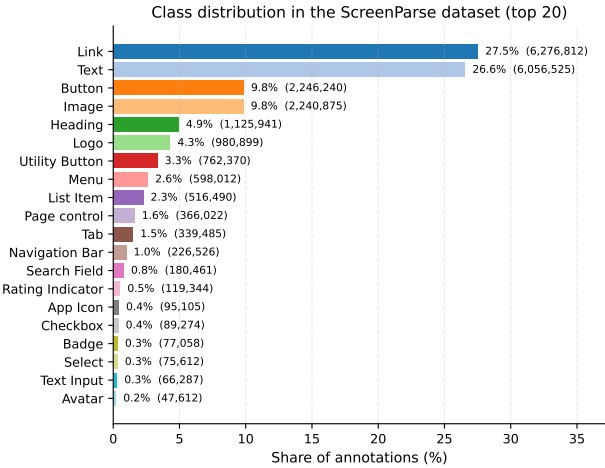

*Figure 1.* Class distribution of the top-20 most frequent UI elements in the *ScreenParse* dataset.

ScreenParse contains **771K** rendered webpage screenshots with **21M** UI element annotations spanning **55** classes. Importantly, it includes both fine-grained atomic elements and semantically meaningful container elements, enabling models to learn hierarchical structure beyond isolated bounding boxes. Each URL is rendered once, and we remove near-duplicate rendered screenshots using perceptual hashing with Hamming distance $< 8$ to reduce visual overlap across splits. We split the dataset into train/val/test using a **90/5/5%** split; Tab. 1 reports split sizes, and Fig. 1 shows the class distribution of the most frequent UI types.

**Comparison to Prior Datasets.** Tab. 2 contrasts Screen-Parse with representative GUI grounding datasets. We mark a dataset as *complete annotation* if it labels (approximately) *all visible UI elements per screen*, rather than only task-relevant or instruction-referred elements. ScreenParse provides complete annotations at substantially larger scale and with a more fine-grained label taxonomy, while preserving hierarchical structure through container elements. This makes ScreenParse particularly suited for pre-training and evaluating models that aim to build holistic screen understanding, and also provides a strong supervision source for training detector-based parsers under a unified taxonomy. Next, we describe Webshot in detail.

### 3.2. Dataset Pipeline: Webshot

ScreenParse is generated entirely by our automated **Webshot** pipeline, which renders diverse URLs, extracts DOM-

*Table 1.* Statistics of the *ScreenParse* dataset.

| Split | Images | Annotations |
|---|---|---|
| Train | 693,975 | 18,968,147 |
| Val | 38,850 | 1,062,552 |
| Test | 38,633 | 1,063,544 |
| Total | 771,458 | 21,094,243 |

*Table 2.* Comparison of grounding datasets. *Complete annotation* indicates whether all visible UI elements on each screen are labeled (dense), as opposed to only a sparse subset (e.g., task-relevant or instruction-referred elements). #E and #S denote the numbers of labeled elements and samples, respectively. [*]These datasets do not define a well-specified set of UI element types.

| Grounding Dataset | Complete annotation | # of types | Scale | |
|---|---|---|---|---|
| | | | # of E | # of S |
| UGround (Gou et al., 2025) | ✗ | 1 | 9M | 773k |
| JEDI (Xie et al., 2024) | ✗ | 4 | 4M | 575k |
| AGUVIS-G (Xu et al., 2025) | ✗ | 1 | 3.8M | 452k |
| OS-ATLAS (Wu et al., 2025) | ✗ | 1[*] | 14.5M | 1.85M |
| RICOSCA (Deka et al., 2017) | ✗ | 1[*] | 170K | 18K |
| UIBert (Bai et al., 2021) | ✗ | 32 | 166K | 57K |
| Widget Caption (Li et al., 2020) | ✗ | 1 | 101K | 14K |
| AMEX (Chai et al., 2025) | ✗ | 2 | 1.2M | 101K |
| ScreenSpot (Cheng et al., 2024) | ✗ | 2 | 3M | 270K |
| GroundCUA (Feizi et al., 2025) | ✓ | 8 | 3.56M | 55k |
| **ScreenParse (Ours)** | ✓ | 55 | 21M | 771k |

aligned candidates, and applies refinement and quality filtering to produce high-coverage dense annotations at scale without human intervention. An overview of the Webshot pipeline is visualized in Fig. 3.

**Web Crawling.** To begin with, we collect a diverse set of web page screenshots by crawling **1 million** unique URLs from the public *45 Million Websites dataset* [1]. This dataset aggregates URLs from multiple sources, including Common Crawl, Alexa Top Sites, and public domain lists. We then curate a balanced subset of URLs spanning various categories (e.g., e-commerce, news, social media, blogs) to ensure diversity in layout and content.

**Annotation Pipeline: Bounding Box Extraction and Filtering.** To obtain dense, screen-complete annotations, we render each URL with Playwright[2] and capture a standardized top-of-page viewport screenshot. For each rendered page, we extract the DOM tree along with associated metadata, then apply cleaning and visibility-based filtering to retain on-screen elements: we remove degenerated boxes and elements with negligible visible area in the rendered viewport, *e.g.,* off-screen/hidden/tiny artifacts, and suppress near-duplicate overlapping boxes introduced by nested DOM wrappers. This yields, per screenshot, bounding boxes, class labels, and text content for all visible UI elements. Crucially, we preserve the DOM hierarchy: in addition to leaf nodes,

---

[1] https://huggingface.co/datasets/Plugiloinc/45_Million_Websites
[2] https://github.com/microsoft/playwright

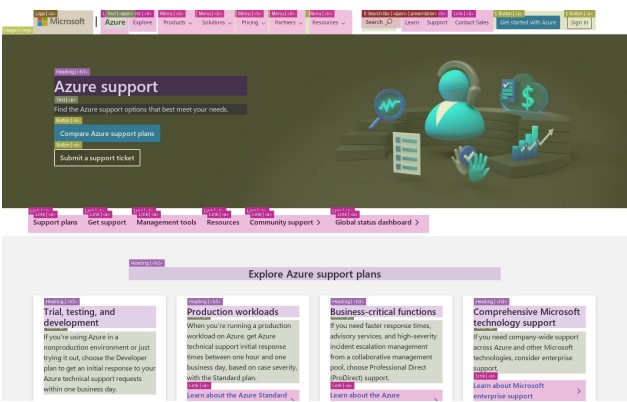

*Figure 2.* Qualitative example from *ScreenParse* illustrating dense, complete UI annotations visualized as labeled bounding boxes.

we annotate enclosing container elements that carry semantic structure *e.g.,* navigation bars, cards, and modals. See Appendix 7.7 for details. Fig. 2 shows an example of the highlighted sample from ScreenParse.

**Annotation Schema.** We defined a taxonomy of **55** UI element classes based on common web design patterns from apple human interface guidelines, Material UI, and Fluent UI design systems (Apple, 2026; MUI, 2026; Microsoft, 2026). The full list of UI element classes is provided in the Appendix Tab. 9.

**Label Refinement and Filtering.** While the heuristic DOM-based labeling step (described above) provides broad coverage, it can be noisy due to heterogeneous and inconsistent markup in real-world web pages. We therefore refine labels with a VLM, using Qwen-3-VL-8B-Instruct (Bai et al., 2025). For each element, we input the rendered screenshot, the element crop, and a compact attribute representation, and prompt the model to predict one of the 55 UI classes. To further suppress noise from dynamic content, ads, and rendering artifacts, we apply a VLM-as-a-judge filter: for each page, we visualize all extracted boxes as an overlay and ask the model to score annotation quality (coverage, false positives, duplicates, and localization). Pages below a quality threshold are discarded. Prompts for both refinement and filtering are provided in Appendix 7.8 and 7.9; additional annotation-quality audit results are provided in Appendix 7.5.

## 4. Method

### 4.1. Problem Formulation

Given a screenshot $I \in \mathbb{R}^{H \times W \times 3}$, screen parsing aims to recover the full set of visible UI elements together with their geometry, semantic type, and text content. Concretely, we represent a screen as a set of elements $S = \{e_i\}_{i=1}^{N}$ where each element $e_i = (b_i, c_i, t_i)$ consists of a bounding box $b_i = (x_1, y_1, x_2, y_2)$, a class label $c_i$ from a fixed UI taxonomy, and optional visible text $t_i$. Unlike single-target grounding, the goal is to predict *all* elements on the screen, including fine-grained widgets and semantically meaningful containers, enabling holistic screen understanding.

### 4.2. ScreenTag: Compact Screen Structure Representation

To train an autoregressive model for dense parsing, we serialize the screen into a compact xml-like structured sequence we call *ScreenTag*, inspired by OTSL (Lysak et al., 2023) and its successor DocTags (Nassar et al., 2025). Each element is emitted as a typed tag followed by discretized location tokens and optional text, and may include its serialized children:

```
<tag> <x1> <y1> <x2> <y2>
[text] [children] </tag>
```

Coordinates are normalized and quantized to a **0–500** grid to balance spatial precision and vocabulary size. This representation is compact and unambiguous to parse from the model output, and it aligns naturally with autoregressive decoding for dense screen parsing.

### 4.3. Lightweight Vision-Language Model

**ScreenVLM.** ScreenVLM is a compact vision–language model that converts a screenshot into a serialized, structured screen representation (*ScreenTag*; Fig. 4). Rather than introducing a heavy new architecture, we adapt a document-to-markup VLM, *Granite Docling*, which is based on the Idefics3 family (Laurençon et al., 2024) and closely related to SmolDocling (Nassar et al., 2025), to the UI domain. Concretely, ScreenVLM couples a SigLIP2-base visual encoder (Tschannen et al., 2025) with a lightweight Granite 165M autoregressive decoder (Mishra et al., 2024). Following the Idefics3 connector design, pixel shuffle with scale factor 4 rearranges 1,024 visual tokens into 64 tokens by grouping 4×4 adjacent patches and concatenating their features (768→12,288), followed by a bias-free linear projection into the 576-dimensional Granite decoder space. This spatial rearrangement preserves local layout information while keeping the visual-token budget small. We initialize from a pretrained Granite Docling checkpoint, since its document conversion pretraining emphasizes localization-aware, structured extraction through markup-like outputs, an inductive bias that transfers naturally to complete screen parsing.

**Training Objective.** A standard sequence cross-entropy treats all tokens equally. However, in screen parsing, *structure* tokens–element types and locations—are far more consequential: small mistakes in tags or box coordinates can invalidate an element even if its text is correct. In addition, text tokens often dominate the sequence length, skewing

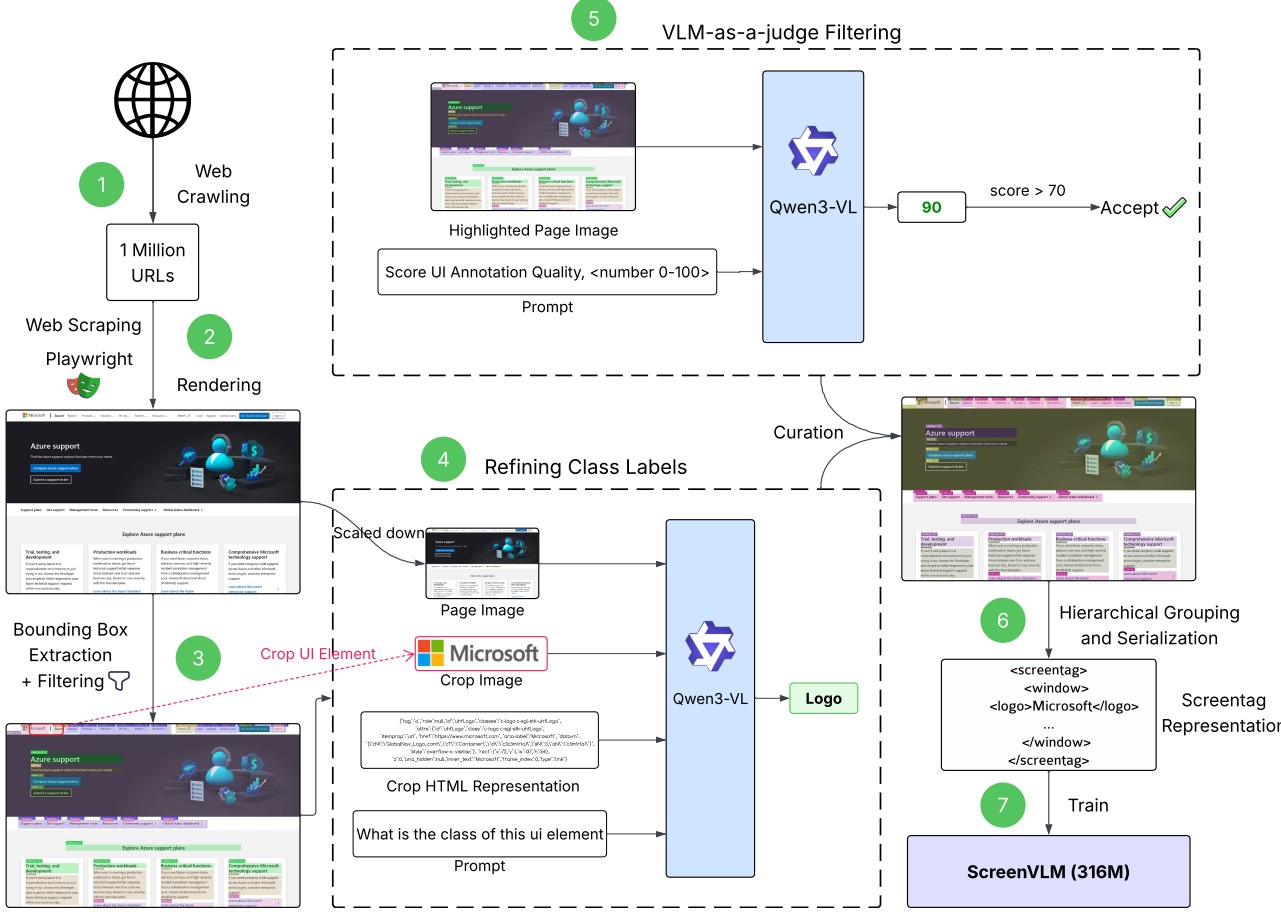

*Figure 3.* Overview of the Webshot dataset generation pipeline. Our scalable framework renders diverse URLs with Playwright and extracts DOM-driven dense annotations. VLMs further refine UI element types and filter low-quality samples using a quality-confidence threshold of 70 on a 0–100 scale.

optimization toward transcription rather than localization and typing. To emphasize structural fidelity, we adopt a structure-aware weighted cross-entropy over the ground-truth ScreenTag sequence:

$$\mathcal{L}(\theta) = -\sum_{t=1}^{T} w(y_t) \log p_\theta(y_t \mid y_{<t}, I),$$

$$w(y_t) = \begin{cases} \lambda_{\text{tag}} & y_t \in \mathcal{V}_{\text{tag}}, \\ \lambda_{\text{loc}} & y_t \in \mathcal{V}_{\text{loc}}, \\ 1 & \text{otherwise}, \end{cases} \quad (1)$$

where $\mathcal{V}_{\text{tag}}$ and $\mathcal{V}_{\text{loc}}$ denote the ScreenTag and location token sets, respectively.

## 5. Experiments

### 5.1. Implementation details.

We fine-tune ScreenVLM on the ScreenParse training split for **287,500** steps using **16** NVIDIA H100 GPUs (2 nodes × 8 GPUs) with an effective batch size of **64**. We use parameter-group learning rates: $2.12 \times 10^{-2}$ for the multi-modal projection (MP) layers and $2 \times 10^{-3}$ for the vision and language backbones. Sequences are truncated or padded to a maximum length of $8192$ tokens. Additional hyperparameters are in Appendix 7.2.

### 5.2. Experimental Setting

We evaluate three questions: (i) whether dense supervision enables accurate *complete screen parsing* in-domain, (ii) whether the resulting perception transfers to public out-of-distribution GUI benchmarks, and (iii) whether our structure-aware loss improves parsing accuracy and transfer.

**Datasets.** We report results on: **ScreenParse** (in-domain dense parsing; *38.6K* test screenshots), **GroundCUA** (Feizi et al., 2025) (OOD GUI screenshots across *87* software platforms; *55K* screenshots, evaluated on the full benchmark), and **ScreenSpot** (Cheng et al., 2024) (sparse grounding across Web/PC/Mobile; *610* test samples total: 199 Web, 210 PC, 201 Mobile). Since these datasets use different label

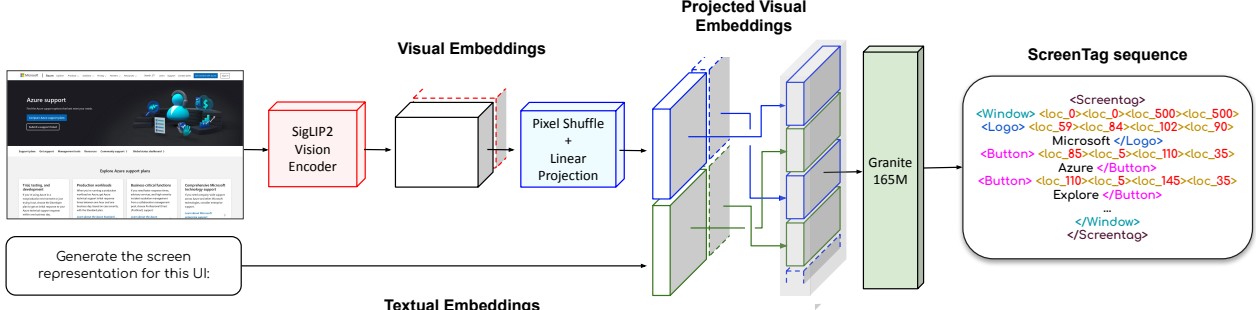

*Figure 4.* Overview of the ScreenVLM architecture. A screenshot is encoded by the SigLIP2 vision encoder (Tschannen et al., 2025) into visual tokens, compressed through pixel shuffle and linear projection, and fed to the Granite-165M LLM (Mishra et al., 2024) decoder together with text tokens to generate the ScreenTag sequence.

spaces, we evaluate with dataset-specific label vocabularies (details in Appendix 7.10).

**Evaluation Metrics.** We measure dense screen parsing quality using *PageIoU* (Niu et al., 2025), which compares the pixel coverage of the union of predicted boxes against the union of ground-truth boxes, capturing how completely a method recovers the screen layout. *Label PageIoU* is the label-aware variant that additionally requires the predicted element type to match the ground truth. We also report *Recall@50*, the fraction of ground-truth elements matched by a prediction with IoU $\geq 0.5$ (and matching class when labels are available), and *mAP@50* for models that provide confidence-ranked detections. For ScreenSpot, which provides sparse target annotations rather than full screens, we report *Recall@50* and *PixCov*, the fraction of annotated target pixels covered by predictions. The formal definitions are given in Appendix 7.3.

**Evaluation protocol.** We evaluate dense parsing on Screen-Parse using PageIoU, Label PageIoU, Recall@50, and (when applicable) mAP@50. For GroundCUA, we evaluate on the full benchmark and report PageIoU/Label PageIoU to measure transfer to real, multi-application UI screenshots. For ScreenSpot (Web/PC/Mobile), we report Recall@50 and PixCov to reflect performance under sparse element annotations. For VLM baselines, we run inference using vLLM (Kwon et al., 2023) with the fixed prompt templates in Appendix 7.10.

### 5.3. Baselines

We compare against two baseline families: (i) foundation VLMs for language-grounded structured extraction and (ii) detector-style UI parsers as strong, efficient localization backbones commonly used in agent pipelines.

**VLM baselines.** We evaluate Qwen3-VL-2B-Instruct, Qwen3-VL-8B-Instruct (Bai et al., 2025), and InternVL3-

2B (Zhu et al., 2025). For each dataset, we prompt models to extract *all* visible UI elements and output bounding boxes, text, and labels constrained to the dataset taxonomy. We use a consistent prompting format across datasets; templates are provided in Appendix 7.10. To quantify the impact of dense ScreenParse supervision on foundation models, we additionally fine-tune InternVL3-2B and Qwen3-VL-2B-Instruct on ScreenParse using ScreenTag format and evaluate them under the same inference protocol.

**Detectors / parsers.** We include OmniParser v2 (Lu et al., 2024), a widely used YOLO-based screen parser. Because it is not trained on our 55-class taxonomy, we primarily report class-agnostic localization metrics for its off-the-shelf outputs (e.g., PageIoU and Recall@50). To evaluate detector-style models under a unified taxonomy and measure how ScreenParse benefits this family, we (i) fine-tune OmniParser v2 on ScreenParse and (ii) train YOLOv11-large (Khanam & Hussain, 2024) and RT-DETRv2 (Lv et al., 2024) on the full 55-class label set.

### 5.4. Experimental Results

We evaluate ScreenVLM and ScreenParse along three axes: (i) in-domain dense parsing on ScreenParse, (ii) out-of-distribution transfer to GroundCUA and ScreenSpot, and (iii) downstream action-task performance. Additional loss-weight sensitivity, annotation analyses, and qualitative examples are provided in the appendix.

**In-domain dense parsing on ScreenParse.** Tab. 3 reports results on the ScreenParse test set. Among VLMs, **ScreenVLM** achieves the strongest dense parsing quality despite being substantially smaller than prompted foundation VLM baselines. Benefiting from our initialization strategy (Sect. 4) and dense ScreenParse supervision, Screen-VLM delivers strong parsing performance and surpasses much larger models. In particular, compared to Qwen3-VL-8B, ScreenVLM improves PageIoU from **0.294** to **0.606**

*Table 3.* ScreenParse test set performance.

| Model | Size | Page IoU | Label Page IoU | mAP@50 |
|---|---|---|---|---|
| *VLMs* | | | | |
| Qwen3-VL-8B-Instruct | 8B | 0.294 | – | – |
| InternVL3-2B | 2B | 0.111 | 0.030 | 0.000 |
| InternVL3-2B + **ScreenParse** | 2B | 0.509 (+0.398) | 0.174 (+0.144) | 0.072 (+0.072) |
| Qwen3-VL-2B-Instruct | 2B | 0.228 | 0.051 | 0.023 |
| Qwen3-VL-2B-Instruct + **ScreenParse** | 2B | 0.585 (+0.357) | 0.166 (+0.115) | 0.152 (+0.129) |
| **ScreenVLM (ours)** | 316M | **0.606** | **0.197** | **0.303** |
| *Detectors / Parsers* | | | | |
| OmniParser V2 | 20M | 0.270 | – | – |
| OmniParser V2 + **ScreenParse** | 20M | 0.503 | 0.141 | 0.251 |
| YOLO + **ScreenParse** | 25M | 0.533 | 0.133 | 0.299 |
| RT-DETRv2 + **ScreenParse** | 43M | **0.600** | **0.172** | **0.362** |

*Table 4.* Performance on the GroundCUA dataset.

| Model | Size | Page IoU | Label Page IoU |
|---|---|---|---|
| *VLMs* | | | |
| Qwen3-VL-8B-Instruct | 8B | 0.060 | 0.010 |
| Qwen3-VL-2B-Instruct | 2B | 0.030 | 0.005 |
| Qwen3-VL-2B-Instruct + **ScreenParse** | 2B | 0.090 (+0.060) | 0.019 (+0.014) |
| InternVL3-2B | 2B | 0.025 | 0.006 |
| InternVL3-2B + **ScreenParse** | 2B | 0.203 (+0.178) | 0.036 (+0.030) |
| **ScreenVLM (ours)** | 316M | **0.251** | **0.043** |
| *Detectors / Parsers* | | | |
| OmniParser V2 | 20M | 0.361 | 0.049 |
| OmniParser V2 + **ScreenParse** | 20M | **0.398** | **0.061** |
| YOLO + **ScreenParse** | 25M | 0.379 | 0.057 |
| RT-DETRv2 + **ScreenParse** | 43M | 0.388 | 0.059 |

and reaches Label PageIoU **0.197**, indicating better global coverage and more accurate label-aware structure recovery. InternVL3-2B and Qwen3-VL-2B perform substantially lower (PageIoU **0.111** and **0.228**, respectively). Overall, these results underscore the value of ScreenParse as dense supervision: prompted VLMs often miss many UI elements, whereas training with complete annotations yields markedly stronger recovery of screen structure. The lower absolute Label PageIoU values reflect the strict 55-class taxonomy, where many errors occur between semantically adjacent classes such as Button, Utility Button, Link, and Text; Appendix 7.5 provides the corresponding confusion analysis.

Detection-based models trained on ScreenParse are also strong. RT-DETRv2 achieves PageIoU **0.600** and mAP@50 **0.362**, while YOLO achieves PageIoU **0.533** and mAP@50 **0.299**, which is expected given their detection-centric architectures. Detector pipelines are strong localizers and can be paired with VLM backends in practical agents, while ScreenVLM directly generates a structured, language-aligned screen state that unifies geometry, semantics, and text. We therefore view VLM and detector approaches as complementary, and report them separately while highlighting that ScreenParse benefits both families.

**Transfer to GroundCUA.** Tab. 4 evaluates transfer to GroundCUA, which contains UI screenshots from diverse applications and is out-of-distribution relative to our web-

only ScreenParse training. Despite this shift, ScreenVLM remains the strongest VLM baseline, achieving PageIoU **0.251** and Label PageIoU **0.043**, compared to Qwen3-VL-8B at **0.060/0.010** and InternVL3-2B at **0.025/0.006**. This indicates that dense parsing supervision induces structural priors that transfer beyond the web domain.

Fine-tuning a foundation VLM on ScreenParse further improves transfer: InternVL3-2B increases from **0.025/0.006** to **0.203/0.036**, narrowing the gap to ScreenVLM and supporting the view that dense screen parsing supervision is broadly beneficial rather than model-specific. Detector-based parsers obtain higher absolute localization scores on GroundCUA (e.g., OmniParser v2 fine-tuned reaches PageIoU **0.398**), consistent with specialization and the evaluation's emphasis on box overlap. Importantly, ScreenVLM reduces the gap to detector pipelines while producing a structured, language-compatible output that is directly usable by downstream agents.

**ScreenSpot: grounding-style evaluation under sparse annotations.** Tab. 5 reports results on ScreenSpot (Web/PC/Mobile). Because ScreenSpot provides only sparse annotations, full-layout metrics such as PageIoU are not directly applicable; we therefore report Recall@50 and PixCov. On the Web split, ScreenVLM achieves Recall@50 **0.557** and PixCov **0.746**, while detector-style parsers attain higher recall (e.g., RT-DETRv2 Recall@50 **0.768**, PixCov **0.857**). On PC and Mobile, ScreenVLM's Recall@50 drops to **0.222** and **0.066**, yet PixCov remains high at **0.839** and **0.847**. This discrepancy suggests that under out-of-distribution UI styles, ScreenVLM often predicts regions that cover annotated pixels (high PixCov) but fails to produce tight element-level boxes required by Recall@50, which is more sensitive to precise localization. Overall, the results indicate non-trivial transfer from ScreenParse supervision, while revealing a clear limitation and future direction: extending dense supervision beyond web pages to better match PC/Mobile UI distributions and improve element-level recall.

**Dense Supervision improves Foundation VLMs.** We further analyze whether ScreenParse benefits models beyond ScreenVLM by fine-tuning multiple foundation VLMs on ScreenParse and comparing against their pretrained counterparts in Tabs. 3, 4, and 5 (see Appendix Tab. 10 for a summary). Across architectures, dense supervision yields consistent gains in both in-domain dense parsing and out-of-distribution transfer. For example, fine-tuning Qwen3-VL-2B improves ScreenParse PageIoU from **0.228** to **0.585**, Label PageIoU from **0.051** to **0.166**, and mAP@50 from **0.023** to **0.152**, and also improves GroundCUA PageIoU from **0.030** to **0.090**. On ScreenSpot, PixCov increases substantially across splits (e.g., Web **0.292**→**0.720**, PC **0.218**→**0.443**), indicating improved grounding under sparse annotations.

*Table 5.* Performance on the ScreenSpot dataset across splits. Numbers under each split indicate # of samples / elements.

| Model | Size | Web | | PC | | Mobile | |
|---|---|---|---|---|---|---|---|
| | | **Recall** | **PixCov** | **Recall** | **PixCov** | **Recall** | **PixCov** |
| | | (199 / 436) | (199 / 436) | (210 / 334) | (210 / 334) | (201 / 502) | (201 / 502) |
| *VLMs* | | | | | | | |
| Qwen3-VL-8B-Instruct | 8B | 0.229 | 0.346 | 0.201 | 0.300 | **0.193** | 0.311 |
| Qwen3-VL-2B-Instruct | 2B | 0.232 | 0.292 | 0.171 | 0.218 | 0.143 | 0.250 |
| Qwen3-VL-2B-Instruct **+ ScreenParse** | 2B | 0.477 (+0.245) | 0.720 (+0.428) | 0.201 (+0.030) | 0.443 (+0.225) | 0.108 (-0.035) | 0.477 (+0.227) |
| InternVL3-2B | 2B | 0.002 | 0.057 | 0.018 | 0.109 | 0.068 | 0.321 |
| InternVL3-2B **+ ScreenParse** | 2B | 0.172 (+0.170) | 0.592 (+0.535) | 0.081 (+0.063) | 0.628 (+0.519) | 0.100 (+0.032) | 0.527 (+0.206) |
| **ScreenVLM (ours)** | 316M | **0.557** | **0.746** | **0.222** | **0.839** | 0.066 | **0.847** |
| *Detectors / Parsers* | | | | | | | |
| OmniParser V2 | 20M | 0.541 | 0.629 | 0.483 | 0.557 | 0.489 | 0.521 |
| OmniParser V2 **+ ScreenParse** | 20M | 0.727 | 0.800 | 0.536 | 0.624 | 0.552 | 0.735 |
| YOLO **+ ScreenParse** | 25M | 0.683 | 0.798 | 0.521 | 0.597 | 0.504 | 0.682 |
| RT-DETRv2 **+ ScreenParse** | 43M | **0.768** | **0.857** | **0.590** | **0.699** | **0.584** | **0.736** |

The same trend holds for a different model family: after fine-tuning, InternVL3-2B improves from **0.111→0.509** PageIoU and **0.036→0.174** Label PageIoU on ScreenParse, and from **0.025→0.203** PageIoU and **0.006→0.036** Label PageIoU on GroundCUA. Together, these results indicate that the benefit of ScreenParse is not model-specific: dense, screen-level supervision consistently strengthens holistic screen understanding and transfers to new UI domains, sometimes enabling a smaller fine-tuned foundation model to outperform a larger prompted counterpart.

### 5.5. Downstream Action-Task Evaluation

**Downstream action-task performance.** To test the effect of better grounding on downstream action tasks, we use an OmniParser-v2-style Set-of-Mark setup with Qwen3-VL-8B-Instruct as the fixed VLM backend, holding OCR, icon captioning, rendering, prompting, and decoding fixed while swapping only the detector. Tab. 6 shows that ScreenParse-trained YOLO improves the primary metric on ScreenSpot, ScreenSpot-Pro, all three official Mind2Web test splits, and OSWorld-G. These are step-level grounding/action-selection evaluations rather than full multi-step agent success, but the controlled setup isolates the perception module and shows that stronger ScreenParse-trained grounding improves the downstream pipeline across web, mobile, and desktop-oriented settings. The OSWorld-G breakdown is especially diagnostic: gains appear across all four capability categories, and the largest improvement is on fine-grained manipulation, where small localization errors are most likely to change the selected action. This suggests that dense screen-level supervision improves not only aggregate recognition, but also the precise target selection needed by GUI action pipelines. Detailed supporting metrics are in Appendix 7.5.

**Structure-aware loss ablation.** Tab. 7 validates the structure-aware training objective introduced in Sec. 4. Compared with standard cross-entropy, upweighting tag and location tokens improves both in-domain dense parsing and out-of-distribution transfer, with especially large gains on the ScreenSpot PC split. This supports the motivation that complete screen parsing should prioritize structural tokens rather than allowing long text spans to dominate the sequence loss.

**Inference efficiency.** Tab. 8 shows that ScreenVLM is substantially more efficient than 2B-parameter foundation VLM baselines under the same vLLM setup. ScreenVLM is about $6.8\times$ smaller than Qwen3-VL-2B-Instruct in model size and achieves 3.62 samples/s, compared with 0.78 samples/s for Qwen3-VL-2B-Instruct and 0.79 samples/s for InternVL3-2B. This supports our design goal of dense screen parsing with a compact model suitable for latency-sensitive computer-use pipelines.

**Takeaway.** Together, these results show that ScreenParse is not tied to one architecture: it improves compact end-to-end parsing, foundation VLM fine-tuning, detector-based parsing, and downstream Set-of-Mark pipelines. This supports dense, screen-complete supervision as a broadly useful signal for GUI understanding.

## 6. Conclusion

We presented **ScreenParse**, a web-scale dataset for *complete* screen parsing with dense UI element annotations (boxes, 55-class types, and text) generated by our automated Webshot pipeline. Using ScreenParse, we trained **ScreenVLM**, a compact VLM that predicts a structured screen state, and introduced a structure-aware objective that emphasizes structure-critical tokens. Across in-domain

*Table 6.* Downstream action-task performance with a fixed Qwen3-VL-8B-Instruct Set-of-Mark pipeline. Only the detector is swapped, isolating the effect of ScreenParse-trained grounding. Values are percentages; gains are percentage points.

| Benchmark / split | Metric | OmniParser v2 | ScreenParse (YOLO) | Δ |
|---|---|---|---|---|
| *Primary benchmark metrics* | | | | |
| ScreenSpot | Action Acc. | 77.8 | **80.1** | +2.3 |
| ScreenSpot-Pro | Action Acc. | 28.5 | **30.5** | +2.0 |
| Mind2Web (website) | Step Success | 24.1 | **27.1** | +3.0 |
| Mind2Web (task) | Step Success | 26.8 | **27.0** | +0.2 |
| Mind2Web (domain) | Step Success | 28.7 | **29.6** | +0.9 |
| OSWorld-G | Overall Acc. | 46.67 | **48.82** | +2.15 |
| *ScreenSpot domain breakdown* | | | | |
| ScreenSpot (Web) | Action Acc. | 77.1 | **81.9** | +4.8 |
| ScreenSpot (Mobile) | Action Acc. | 83.5 | **84.9** | +1.4 |
| ScreenSpot (Desktop) | Action Acc. | 70.4 | **71.2** | +0.8 |
| *OSWorld-G capability breakdown* | | | | |
| OSWorld-G (Text Matching) | Acc. | 59.00 | **61.69** | +2.69 |
| OSWorld-G (Element Recognition) | Acc. | 50.30 | **50.91** | +0.61 |
| OSWorld-G (Layout Understanding) | Acc. | 53.36 | **56.13** | +2.77 |
| OSWorld-G (Fine-grained Manipulation) | Acc. | 25.50 | **29.53** | +4.03 |

*Table 7.* Ablation on ScreenVLM training loss. We compare standard cross-entropy (CE) against our structure-aware weighted loss on ScreenParse, GroundCUA, and ScreenSpot (Web/PC/Mobile). Recall denotes class-agnostic Recall@50.

| Model | ScreenParse | | GroundCUA | | ScreenSpot (Web) | ScreenSpot (PC) | ScreenSpot (Mobile) |
|---|---|---|---|---|---|---|---|
| | Page IoU | Label Page IoU | Page IoU | Label Page IoU | Recall | Recall | Recall |
| ScreenVLM (CE) | 0.592 | 0.192 | 0.226 | 0.039 | 0.541 | 0.129 | 0.052 |
| ScreenVLM (Structure-aware) | **0.606** (+2.4%) | **0.197** (+2.6%) | **0.251** (+11.1%) | **0.043** (+10.3%) | **0.557** (+3.0%) | **0.222** (+72.1%) | **0.066** (+26.9%) |

*Table 8.* Inference efficiency with vLLM on one H100. Latency is mean ± std over 128 samples.

| Model | Size (MB) | Latency (ms)↓ | Throughput (s⁻¹)↑ |
|---|---|---|---|
| Qwen3-VL-2B-Instruct | 4300 | 1289.1 ± 251.7 | 0.78 |
| InternVL3-2B | 4178 | 1267.3 ± 187.9 | 0.79 |
| **ScreenVLM (ours)** | **632** | **276.4 ± 139.0** | **3.62** |

evaluation on ScreenParse and transfer to public benchmarks (GroundCUA and ScreenSpot) (Feizi et al., 2025; Cheng et al., 2024), ScreenVLM substantially outperforms prompted foundation VLM baselines, while fine-tuning multiple foundation VLMs and parsers on ScreenParse yields consistent improvements, supporting the value of dense screen-level supervision for holistic UI understanding.

**Limitations.** While models trained on ScreenParse show consistent transfer beyond web pages, ScreenParse is predominantly web-centric, leaving a domain gap to native desktop/mobile applications and UI toolkits. The gap is most visible in tight element-level localization on PC/mobile screens, even when pixel coverage remains high. Moreover, although Webshot applies extensive filtering and refinement, DOM-driven extraction can still contain residual noise from dynamic content, canvas-heavy interfaces, ads, overlays, and rendering artifacts, which may affect a subset of annotations. Finally, ScreenParse is built from viewport-level captures rather than scroll-complete page states. This keeps each sample tied to a well-defined visible screen and avoids noise from lazy loading or scroll-triggered layout changes, but it underrepresents below-the-fold elements and long-range scrolling contexts.

**Future Work.** A natural next step is to expand dense parsing supervision beyond web pages to cover native desktop/mobile UIs and richer interaction contexts. Another promising direction is to leverage screen-parsing-pretrained models as strong visual backbones for *vision-language-action* agents by fine-tuning them on downstream interaction tasks (e.g., click/type/scroll). This would capitalize on the holistic, language-aligned screen state to improve grounding and decision making.

## Impact Statement

ScreenParse and ScreenVLM aim to improve screen understanding for computer-use agents, assistive technologies, and automated UI testing. More reliable dense UI parsing can support accessibility tools for users who rely on screen interpretation, and can make GUI agents more robust by exposing a structured view of visible elements. At the same time, stronger GUI perception could be misused for unauthorized automation, large-scale scraping, or deceptive UI interaction. We mitigate these risks by constructing ScreenParse from publicly available web pages, avoiding personally identifiable information where possible, and planning research-oriented release guidelines for the dataset, model, and code.

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

*Table 9.* ScreenTag screen parsing classes (55 total) used in our dataset generation and screen parsing experiments.

| ID | Class | ID | Class | ID | Class |
|---|---|---|---|---|---|
| 1 | Table | 20 | Scroll | 39 | Bottom navigation |
| 2 | Column/Browser | 21 | Switch | 40 | Breadcrumb |
| 3 | Button | 22 | File Icon | 41 | Page control |
| 4 | Utility Button | 23 | Chart | 42 | Link |
| 5 | App Icon | 24 | Window | 43 | Menu |
| 6 | Navigation Bar | 25 | Screen | 44 | Pagination |
| 7 | Status Bar | 26 | List | 45 | Tab |
| 8 | Search Field | 27 | List Item | 46 | Search Bar |
| 9 | Toolbar | 28 | PopUp Menu | 47 | Date-Time picker |
| 10 | Tooltip | 29 | Steppers | 48 | Calendar |
| 11 | Video | 30 | Toggles | 49 | Text |
| 12 | Tab Bar | 31 | Text Input | 50 | Heading |
| 13 | Side Bar | 32 | Rating Indicator | 51 | Code snippet |
| 14 | Slider | 33 | Checkbox | 52 | Carousel |
| 15 | Picker | 34 | Radiobox | 53 | Notification |
| 16 | ContextMenu | 35 | Select | 54 | Logo |
| 17 | DockMenu | 36 | Avatar | 55 | Progress bar |
| 18 | EditMenu | 37 | Badge | | |
| 19 | Image | 38 | Alert | | |

# 7. Appendix

## 7.1. Screen Parsing Label Set (ScreenTag)

Tab. 9 lists the 55 semantic classes used for screen parsing in our ScreenTag annotation schema.

## 7.2. Training Details

**Qwen3-VL-2B-Instruct Finetuning.** We fine-tune Qwen3-VL-2B-Instruct on ScreenParse with BF16 and DeepSpeed ZeRO-3 offload, updating only the multimodal LLM (vision tower and projector frozen). We use batch size 1 with gradient accumulation 4, using AdamW optimizer with cosine schedule (3% warmup), learning rate $2 \times 10^{-5}$, weight decay 0.01, max sequence length 8192, and train for 5 epochs.

**InternVL3-2B Finetuning.** We fine-tune InternVL3-2B on ScreenParse using BF16 and DeepSpeed (ZeRO stage 1), freezing the vision backbone while updating the LLM and MLP. We train for 5 epochs with total batch size 128 (8 GPUs, batch size 4 per GPU, gradient accumulation 4), AdamW with cosine schedule (3% warmup), learning rate $2 \times 10^{-5}$, weight decay 0.05, and max sequence length 8192 with gradient checkpointing.

**RT-DETRv2 Training.** We train RT-DETRv2 with a PResNet-50-VD backbone, HybridEncoder, and RTDETRTransformerv2 on our COCO-format dataset (55 classes). Training uses total batch size 128 (val 64), images resized to $736 \times 1280$, and augmentations including photometric distort, zoom-out, IoU crop, and random horizontal flip; heavy augmentations and multiscale are disabled after epoch 71. We optimize with AdamW (lr $8 \times 10^{-4}$, betas 0.9/0.999, weight decay $1 \times 10^{-4}$), using a lower backbone LR ($8 \times 10^{-5}$) and zero weight decay for encoder/decoder norm/bn. We train for 72 epochs with linear warmup for 2000 iterations and a MultiStepLR (milestone 1000, gamma 0.1), and clip gradients at 0.1.

**YOLOv11-L Training.** We train a YOLOv11-L on the ScreenParse for 500 epochs at 1280 resolution with batch size 48 on 8 GPUs. We use AdamW with cosine learning-rate schedule (`lr0` $= 2.08 \times 10^{-4}$, `lrf` $= 0.05$), momentum 0.9, weight decay $5 \times 10^{-4}$, and a short warmup ($\approx$0.5 epochs) with patience of 25 epochs. We disable heavy composition augmentations (mosaic/mixup) and instead use mild geometric/color jitter with multi-scale training.

**OmniParser v2 Finetuning.** We fine-tune OmniParser v2 using the same training setup and augmentation configuration as our YOLOv11-L training, changing only the optimization schedule: we train for 100 epochs (vs. 500), use a higher base learning rate `lr0` $= 2 \times 10^{-3}$ (vs. $2.08 \times 10^{-4}$), and set early-stopping patience to 20 (vs. 25).

For the structure-aware loss in Eq. 1 we set $\lambda_{\text{tag}} = 2$ and $\lambda_{\text{loc}} = 2$.

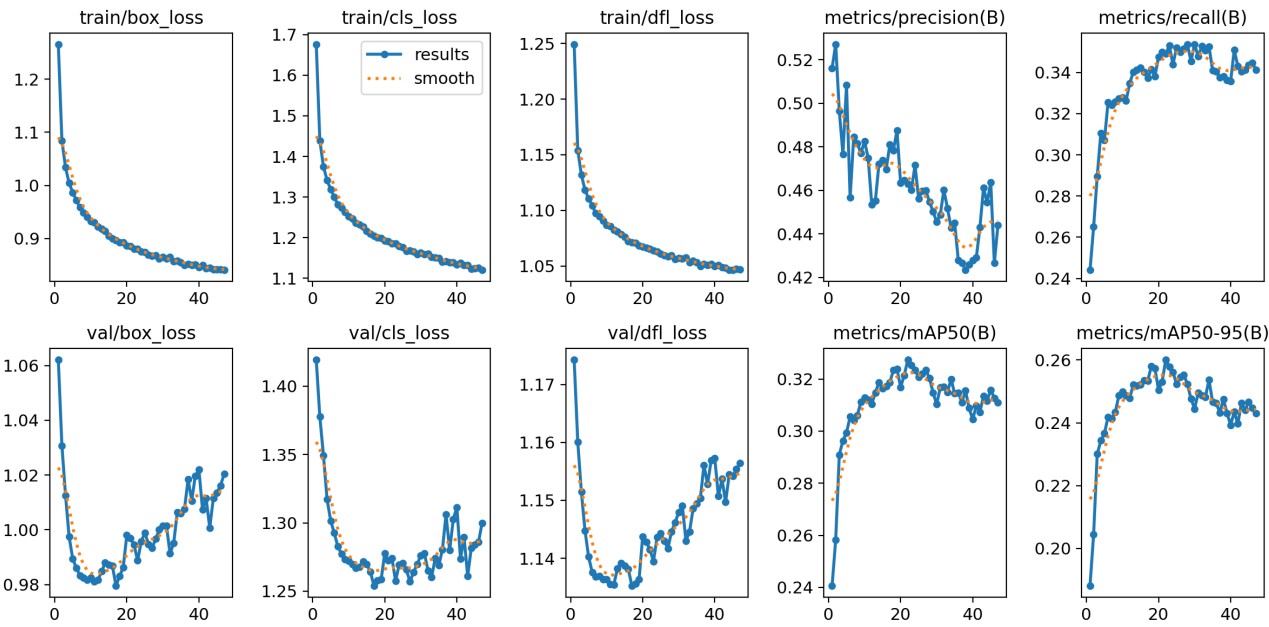

*Figure 5.* Training/Validation loss and accuracy curves for the YOLO component.

## 7.3. Evaluation Metrics

We use the indicator function $\mathbf{1}[\cdot]$, defined as $\mathbf{1}[s] = 1$ if statement $s$ is true and $0$ otherwise.

Let $G$ be the set of ground-truth boxes and $P$ the set of predicted boxes for an image with pixel domain $\Omega$.

**PageIoU.** We define occupancy masks over pixels:

$$M_G(p) = \mathbf{1}[\exists\, g \in G \text{ s.t. } p \in g], \tag{2}$$

$$M_P(p) = \mathbf{1}[\exists\, b \in P \text{ s.t. } p \in b]. \tag{3}$$

PageIoU measures layout-level overlap between the unions of boxes:

$$\text{PageIoU}(P, G) = \frac{\sum_{p \in \Omega} M_P(p)\, M_G(p)}{\sum_{p \in \Omega} \mathbf{1}[M_P(p) + M_G(p) > 0]}. \tag{4}$$

**Label PageIoU.** Let $c(g)$ and $c(b)$ be class labels. We build pixel-wise label maps $L_G(p)$ and $L_P(p)$ by assigning each pixel the label of the *smallest-area* box covering it (background otherwise). Label PageIoU counts intersection only when labels agree:

$$\text{LabelPageIoU}(P, G) = \frac{\sum_{p \in \Omega} \mathbf{1}[L_P(p) = L_G(p) \ \wedge\ L_G(p) \neq \text{bg}]}{\sum_{p \in \Omega} \mathbf{1}[M_P(p) + M_G(p) > 0]}. \tag{5}$$

**Recall@50.** Let $\text{IoU}(b, g) = \frac{|b \cap g|}{|b \cup g|}$. A ground-truth box $g$ is matched if there exists a prediction $b$ with $\text{IoU}(b, g) \geq 0.5$ (and, for label-aware recall, $c(b) = c(g)$). We compute one-to-one matches greedily by prediction confidence. Then

$$\text{Recall@50} = \frac{1}{|G|} \sum_{g \in G} \mathbf{1}[g \text{ is matched}]. \tag{6}$$

**PixCov (pixel coverage).** For datasets with sparse target annotations (e.g., ScreenSpot), we report pixel coverage of the annotated target area:

$$\text{PixCov}(P, G) = \frac{\sum_{p \in \Omega} M_P(p) \, M_G(p)}{\sum_{p \in \Omega} M_G(p)}. \tag{7}$$

**mAP@50.** We compute label-aware mAP@50 with one-to-one greedy matching at IoU $\geq 0.5$ (same-class), rank predictions by confidence when available (otherwise use score $= 1.0$), and average AP over classes present in the ground truth:

$$\text{mAP@50} = \frac{1}{|\mathcal{C}^+|} \sum_{k \in \mathcal{C}^+} \text{AP}_k @50, \tag{8}$$

where $\mathcal{C}^+ = \{k \in \mathcal{C} \mid |G_k| > 0\}$ denotes classes that appear in the ground truth.

## 7.4. Additional Results

This section includes supplementary tables and ablations referenced in the main paper.

*Table 10.* Ablation on foundation VLMs finetuning with ScreenParse. We report results on ScreenParse, GroundCUA, and ScreenSpot (Web/PC/Mobile splits). Recall denotes class-agnostic Recall@50 and PixCov denotes PageIoU recall.

| Model | Size | ScreenParse | | | GroundCUA | | ScreenSpot (Web) | | ScreenSpot (PC) | | ScreenSpot (Mobile) | |
|---|---|---|---|---|---|---|---|---|---|---|---|---|
| | | Page IoU | Label Page IoU | mAP@50 | Page IoU | Label Page IoU | Recall | PixCov | Recall | PixCov | Recall | PixCov |
| InternVL3-2B | 2B | 0.116 | 0.030 | 0.000 | 0.025 | 0.006 | 0.002 | 0.057 | 0.018 | 0.109 | 0.068 | 0.321 |
| InternVL3-2B + ScreenParse | 2B | **0.497** (+0.381) | **0.155** (+0.125) | **0.063** (+0.063) | **0.203** (+0.178) | **0.036** (+0.030) | **0.172** (+0.170) | **0.592** (+0.535) | **0.081** (+0.063) | **0.628** (+0.519) | **0.100** (+0.032) | **0.527** (+0.206) |
| Qwen3-VL-2B-Instruct | 2B | 0.228 | 0.051 | 0.023 | 0.030 | 0.005 | 0.232 | 0.292 | 0.171 | 0.218 | 0.143 | 0.250 |
| Qwen3-VL-8B-Instruct | 8B | 0.281 | 0.043 | 0.000 | 0.060 | 0.010 | 0.229 | 0.346 | 0.201 | 0.300 | **0.193** | 0.311 |
| Qwen3-VL-2B-Instruct + ScreenParse | 2B | **0.585** (+0.357) | **0.166** (+0.115) | **0.152** (+0.129) | **0.090** (+0.060) | **0.019** (+0.014) | **0.477** (+0.245) | **0.720** (+0.428) | **0.201** (+0.030) | **0.443** (+0.225) | 0.108 (-0.035) | **0.477** (+0.227) |

## 7.5. Additional Dataset and Downstream Analyses

This section provides the detailed analyses summarized in the main paper, including the downstream detector-swap study, annotation audit, label-confusion analysis, and loss ablations.

*Table 11.* Detailed downstream action-task evaluation under a fixed Set-of-Mark pipeline. We keep OCR, icon captioning, SoM rendering, prompting, and the Qwen3-VL-8B-Instruct backend fixed, and swap only the detector. *ScreenParse (YOLO)* denotes our ScreenParse-trained detector. All numbers are percentages; $\Delta$ is reported in percentage points.

| Benchmark / split | Metric | OmniParser v2 | ScreenParse (YOLO) | $\Delta$ |
|---|---|---|---|---|
| *Primary benchmark metrics* | | | | |
| ScreenSpot | Action Acc. | 77.8 | **80.1** | +2.3 |
| ScreenSpot-Pro | Action Acc. | 28.5 | **30.5** | +2.0 |
| Mind2Web (website) | Step Success | 24.1 | **27.1** | +3.0 |
| Mind2Web (task) | Step Success | 26.8 | **27.0** | +0.2 |
| Mind2Web (domain) | Step Success | 28.7 | **29.6** | +0.9 |
| *ScreenSpot domain breakdown* | | | | |
| ScreenSpot (Web) | Action Acc. | 77.1 | **81.9** | +4.8 |
| ScreenSpot (Mobile) | Action Acc. | 83.5 | **84.9** | +1.4 |
| ScreenSpot (Desktop) | Action Acc. | 70.4 | **71.2** | +0.8 |
| *Supporting metrics* | | | | |
| ScreenSpot | Text Acc. | 83.8 | **87.1** | +3.3 |
| ScreenSpot | Icon Acc. | 70.6 | **71.1** | +0.5 |
| Mind2Web (website) | Element Acc. | 31.1 | **33.9** | +2.8 |
| Mind2Web (website) | Action F1 | 80.7 | **81.1** | +0.4 |
| Mind2Web (task) | Element Acc. | 31.6 | **32.3** | +0.7 |
| Mind2Web (task) | Action F1 | 84.2 | **84.4** | +0.2 |
| Mind2Web (domain) | Element Acc. | 34.2 | **34.6** | +0.4 |
| Mind2Web (domain) | Action F1 | 83.5 | **83.7** | +0.2 |

*Table 12.* OSWorld-G evaluation under the same controlled detector-swap setup. All numbers are percentages; $\Delta$ is the difference between the two detector performances.

| Detector | Text Matching | Element Recognition | Layout Understanding | Fine-grained Manipulation | Overall |
|---|---|---|---|---|---|
| OmniParser v2 | 59.00 | 50.30 | 53.36 | 25.50 | 46.67 |
| ScreenParse (YOLO) | **61.69** | **50.91** | **56.13** | **29.53** | **48.82** |
| $\Delta$ | +2.69 | +0.61 | +2.77 | +4.03 | +2.15 |

*Table 13.* Human audit of ScreenParse annotation quality on 100 randomly sampled pages. The audit evaluates class-label correctness for extracted bounding boxes against the 55-class taxonomy.

| Audit statistic | Value |
|---|---|
| Audited pages | 100 |
| Audited UI elements | 3,173 |
| Correct labels | 3,101 / 3,173 |
| Element-level micro accuracy | 97.73% |
| Mean per-page accuracy | 96.77% |
| Micro F1 | 0.9888 |
| Macro F1 | 0.9208 |
| Pages with $\geq$95% accuracy | 84 / 100 |
| Pages with $\geq$90% accuracy | 94 / 100 |

*Table 14.* Most frequent label confusions after successful localization. Rates are reported per 1,000 ground-truth elements and aggregated over four evaluated models.

| ScreenParse | | GroundCUA | |
|---|---|---|---|
| Confusion pair | Rate / 1K | Confusion pair | Rate / 1K |
| Utility Button $\rightarrow$ Button | 116 | Menu $\rightarrow$ Navigation | 598 |
| Button $\rightarrow$ Link | 67 | Input Element $\rightarrow$ Info Display | 300 |
| Text $\rightarrow$ Link | 52 | Navigation $\rightarrow$ Info Display | 299 |
| Link $\rightarrow$ Text | 35 | Navigation $\rightarrow$ Button | 250 |
| Button $\rightarrow$ Utility Button | 34 | Input Element $\rightarrow$ Button | 225 |
| Link $\rightarrow$ Button | 25 | Menu $\rightarrow$ Info Display | 200 |

*Table 15.* Sensitivity analysis for the structure-aware loss weight. We set $\lambda_{\text{tag}} = \lambda_{\text{loc}} = \lambda$ and report validation loss, PageIoU on ScreenParse (SP) and GroundCUA (GCUA), and Recall@50 on ScreenSpot (SS). The dagger marks the setting used in the submitted paper.

| $\lambda$ | Val. Loss | SP PageIoU | GCUA PageIoU | SS-Web | SS-PC | SS-Mobile |
|---|---|---|---|---|---|---|
| 0.5 | 0.983 | 0.556 | 0.210 | 0.390 | 0.162 | 0.044 |
| 1.0 | 0.943 | 0.563 | 0.236 | 0.456 | 0.162 | 0.040 |
| 1.5 | 0.921 | 0.568 | 0.236 | 0.477 | 0.153 | **0.072** |
| 2.0$^\dagger$ | 0.904 | 0.565 | **0.242** | 0.493 | 0.132 | 0.040 |
| 3.0 | **0.875** | **0.571** | 0.238 | **0.567** | **0.183** | 0.056 |
| 4.0 | 1.407 | 0.462 | 0.233 | 0.039 | 0.018 | 0.028 |

The sensitivity results show that performance improves as structure-critical tags and location tokens are upweighted from $\lambda = 0.5$ to $\lambda = 3.0$, while $\lambda = 4.0$ destabilizes training. The chosen setting $\lambda = 2.0$ is not tuned to maximize every metric, but lies in the stable high-performing regime and achieves the best GroundCUA PageIoU among the tested values.

## 7.6. Qualitative Results

| Ground Truth | Qwen3-VL-8B-Instruct | ScreenVLM (Ours) |
|---|---|---|

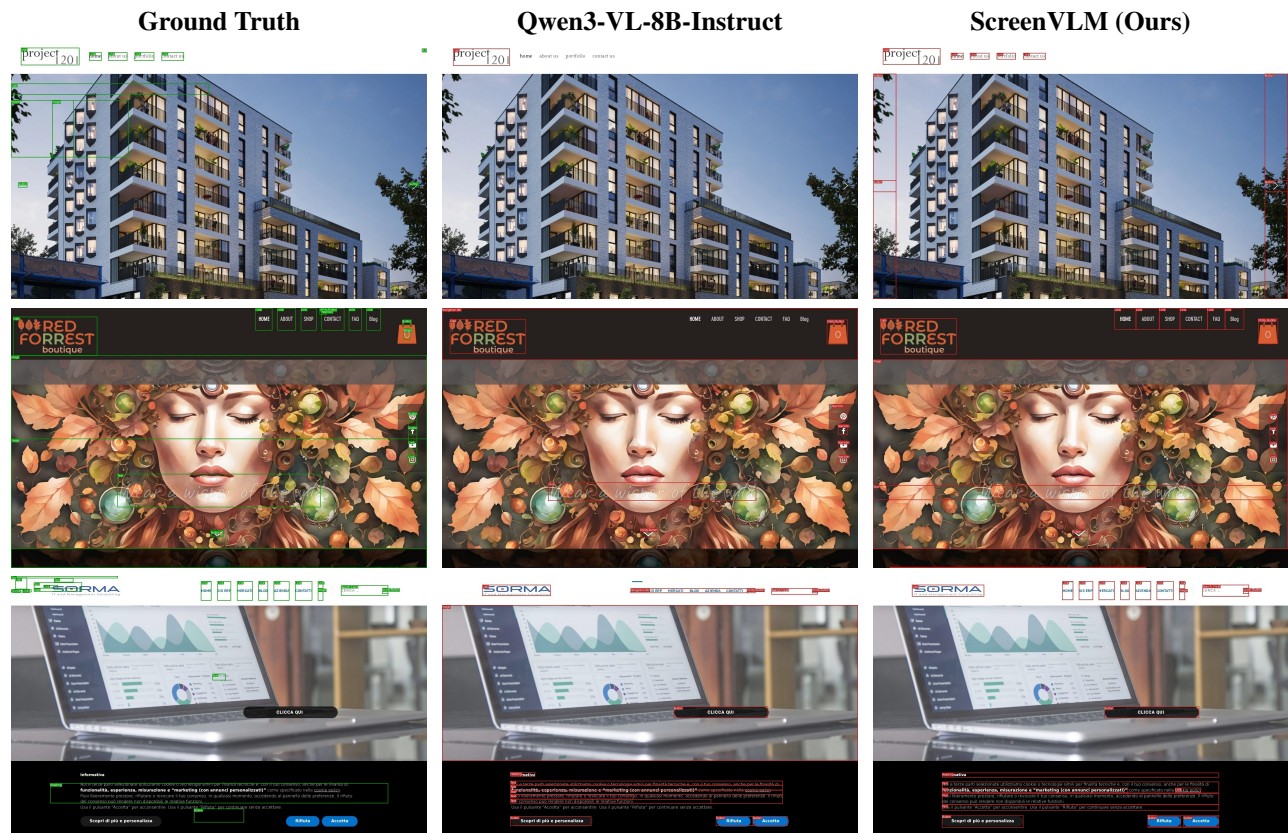

*Figure 6.* Qualitative screen parsing predictions for VLMs. Each row shows the same screenshot across columns; bounding boxes and labels are rendered as overlays. As it can be seen, in terms of recall, localization and granularity of the predictions, our ScreenVLM model outperforms the Qwen3-VL-8B-Instruct model significantly. Some of the ground truth annotations contain errors due to the rendering or DOM extraction issues.

| Ground Truth | OmniParser v2 | YOLO (Ours) |
| --- | --- | --- |

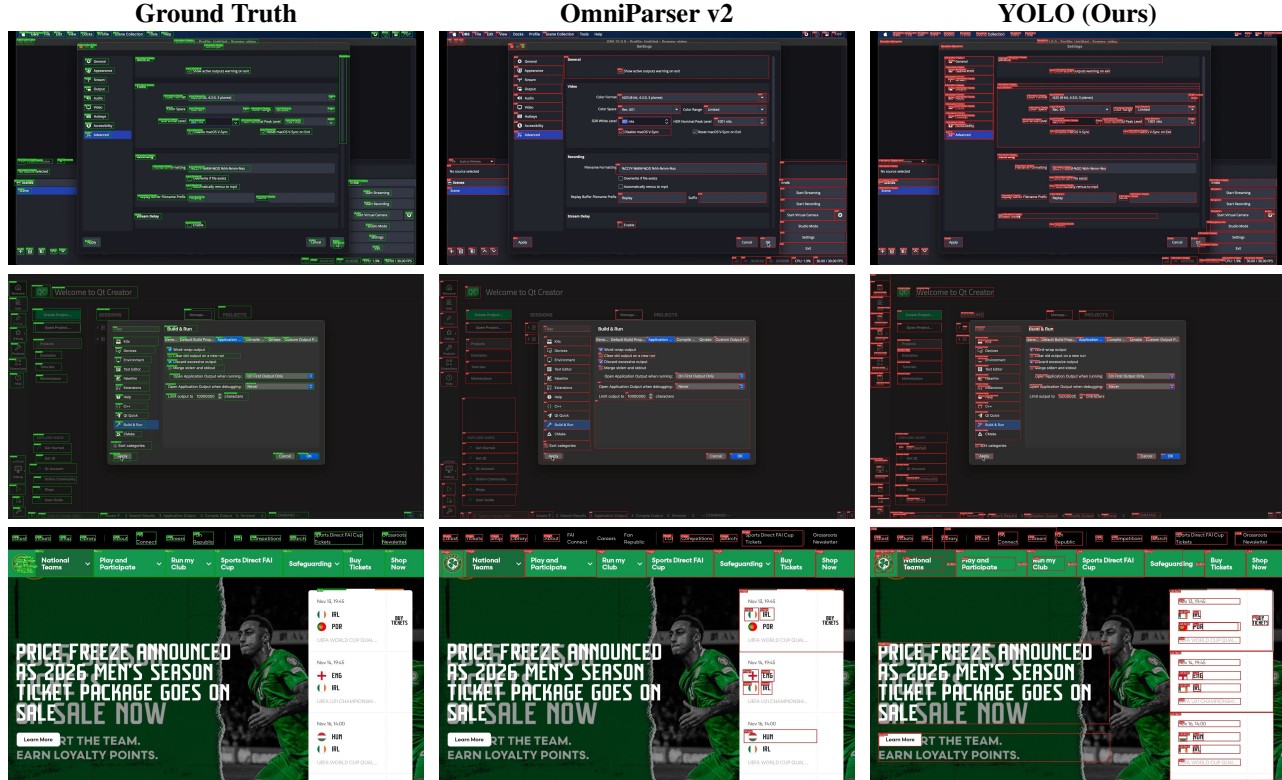

*Figure 7.* Qualitative screen parsing predictions for detector/parser baselines on GroundCUA dataset. Each row shows the same screenshot across columns. Our YOLO model has much less false negatives compared to OmniParser v2, and it covers text areas that may be important for understanding the UI.

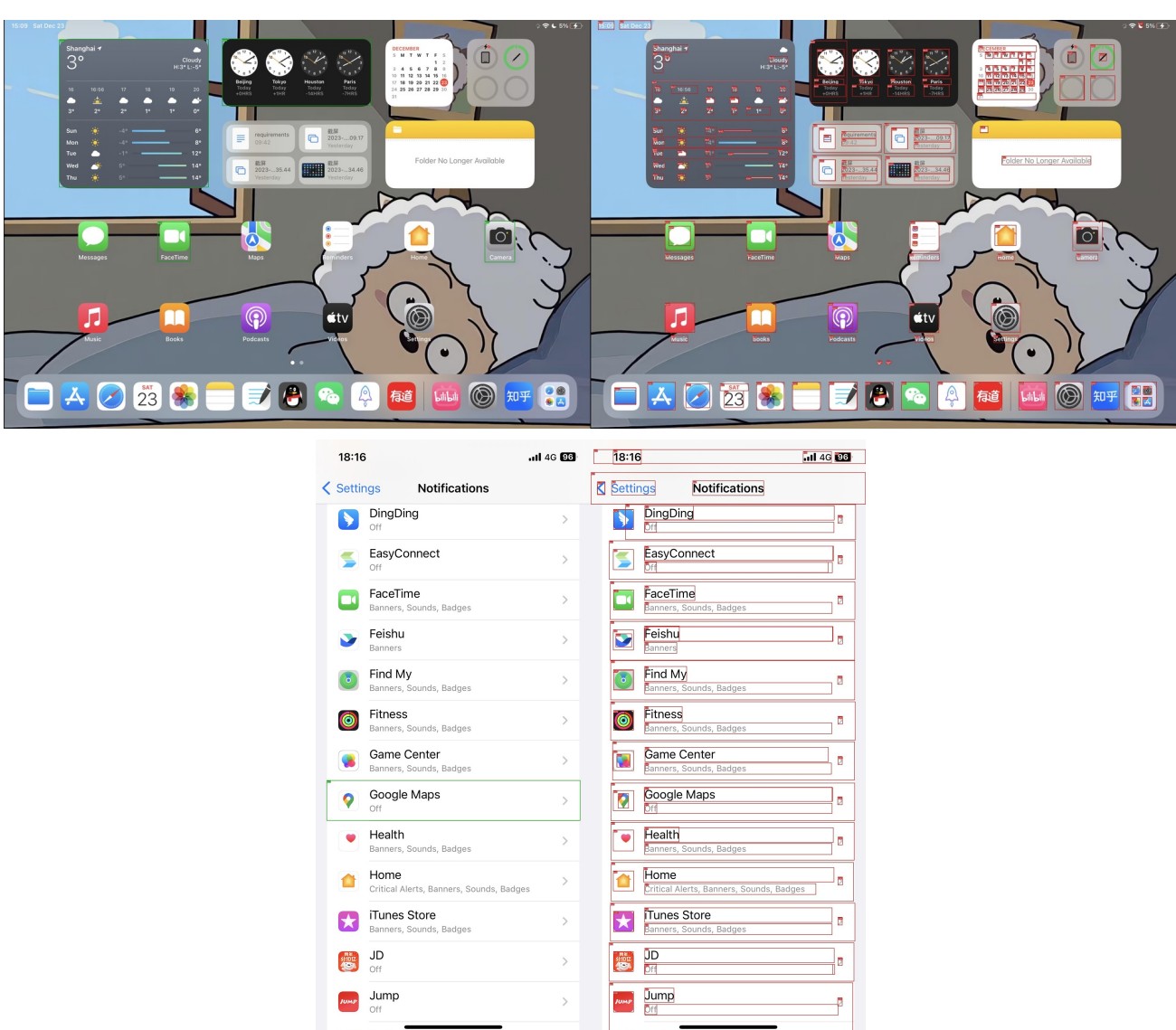

Figure 8. Out-of-distribution qualitative results of our YOLO model on the ScreenSpot *Mobile* split. Each visualization shows ground truth (left) and the model prediction (right). The ground truth visualization is not complete since ScreenSpot provides sparse annotations.

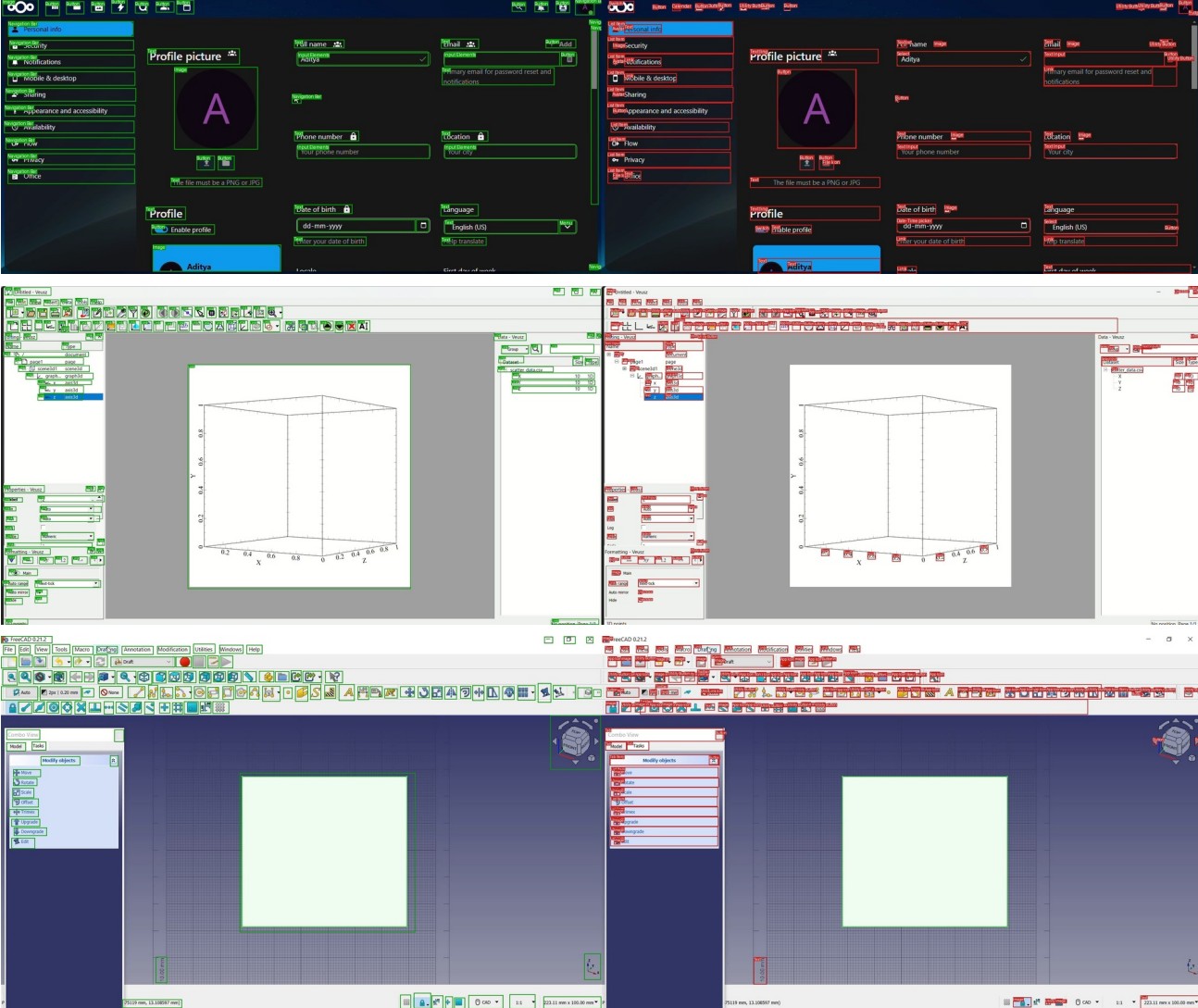

*Figure 9.* Out-of-distribution qualitative results of our YOLO model on the GroundCUA dataset. Each visualization shows ground truth (left) and the model prediction (right).

**Ground Truth**     **OmniParser v2**     **OmniParser v2 + ScreenParse**

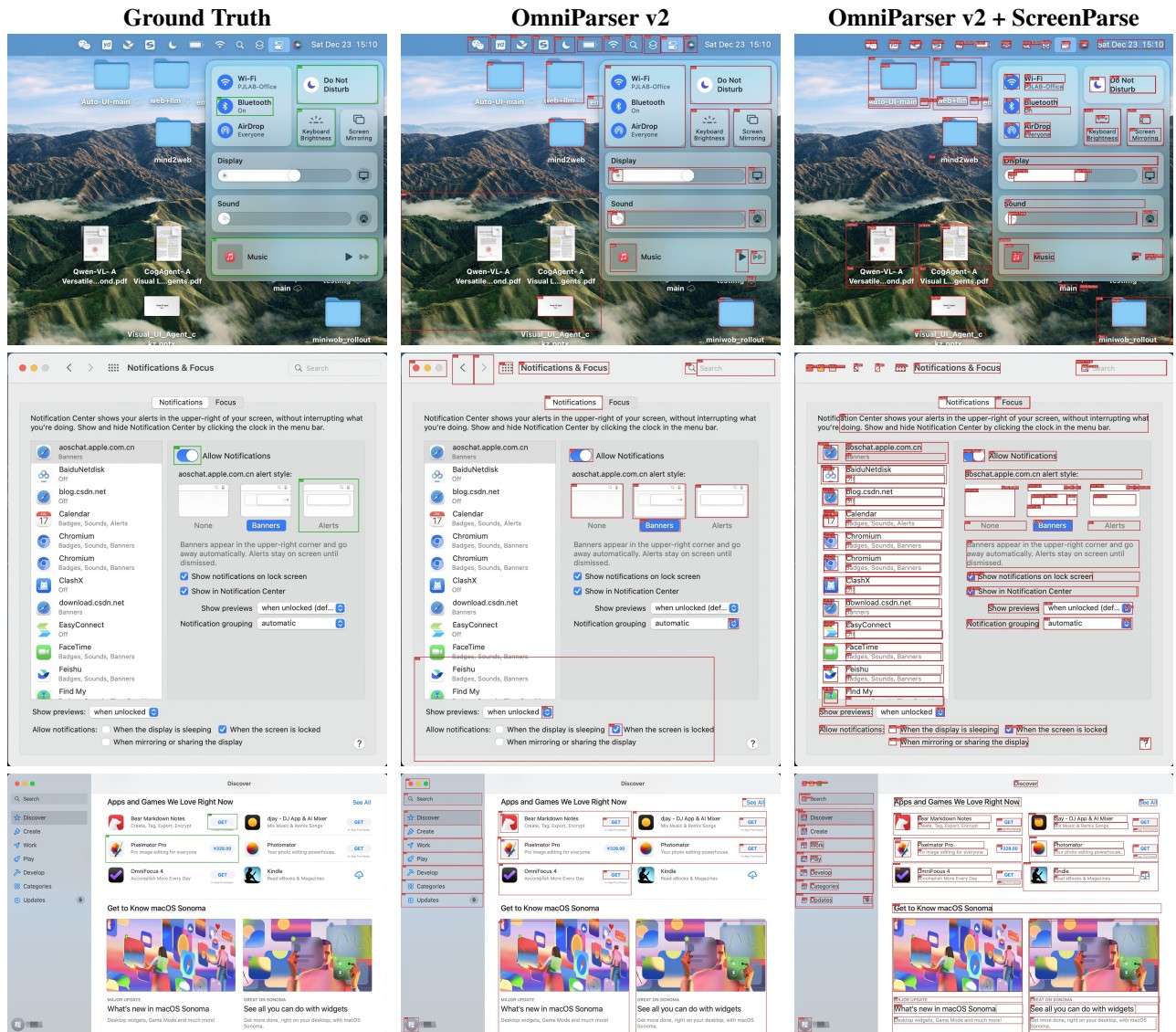

*Figure 10.* Additional qualitative results on ScreenSpot (PC). We compare OmniParser v2 against OmniParser v2 fine-tuned on ScreenParse.

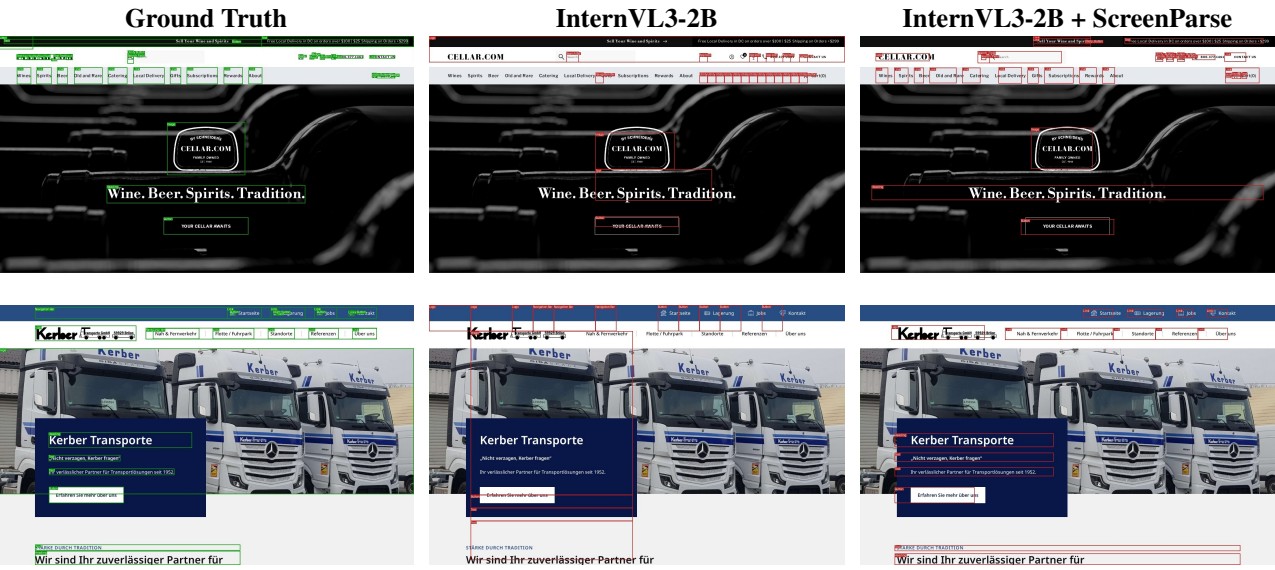

*Figure 11.* Additional qualitative result on ScreenSpot (Mobile): OmniParser v2 vs. OmniParser v2 fine-tuned on ScreenParse.

*Figure 12.* Additional qualitative results on ScreenParse: InternVL3-2B before and after fine-tuning on ScreenParse.

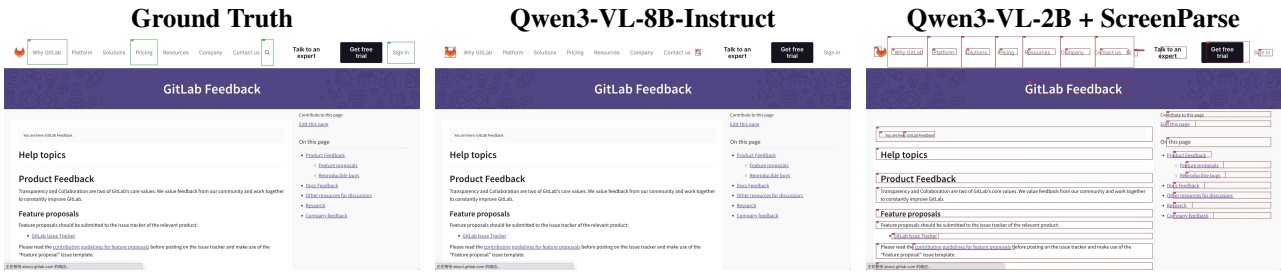

*Figure 13.* Additional qualitative result on ScreenSpot (Web): prompted Qwen3-VL-8B vs. Qwen3-VL-2B fine-tuned on ScreenParse.

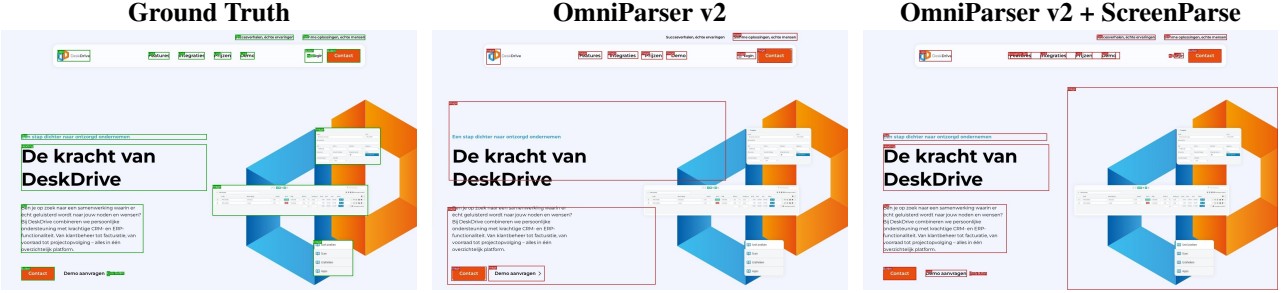

*Figure 14.* Additional qualitative result on ScreenParse: OmniParser v2 before and after fine-tuning on ScreenParse.

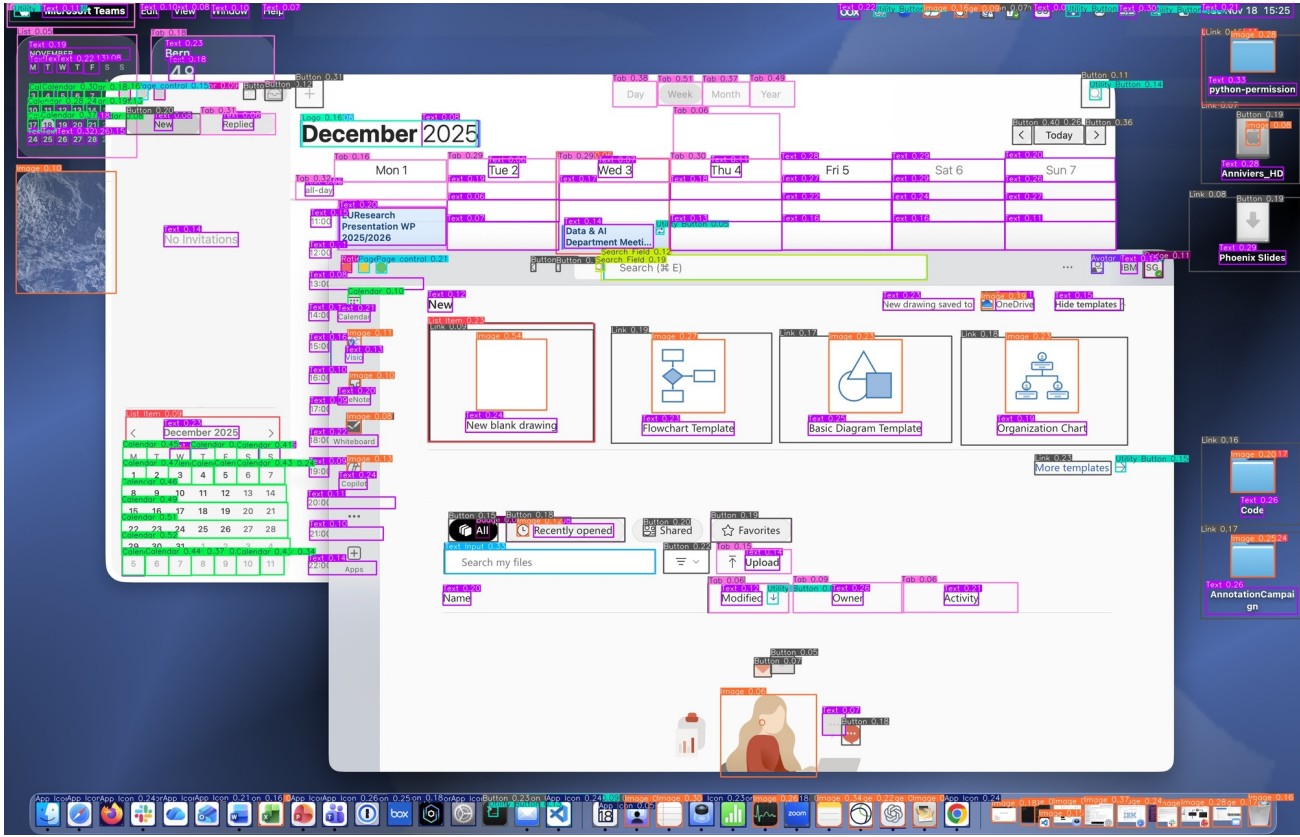

*Figure 15.* Out-of-distribution qualitative result of our YOLO detector on a complex desktop multi-window screen.

## 7.7. Webshot Pipeline Specifications

### 7.7.1. RENDERING AND SCREENSHOT STANDARDIZATION

**Controlled rendering setup.** All pages are rendered in a standardized browser environment (Chromium via Playwright) with viewport size of 1440 width and 900 height. We disabled CSS animations/transitions to avoid transient states and inconsistent layouts across runs. We use a bounded navigation timeout of **30s** and allow a short post-load settling period (**800ms**) before extracting annotations.

**Viewport-only capture (default).** By default, we capture the *top-of-page viewport* without scrolling. This makes the definition of "visible UI elements" unambiguous and avoids mixing content from multiple scroll positions into a single example. A full-page mode (with controlled scrolling to trigger lazy-loading) is supported, but ScreenParse is constructed under viewport-only capture for consistency.

### 7.7.2. DENSE UI ELEMENT EXTRACTION WITH DOM HIERARCHY

**Element set and geometry.** For each rendered page, we extract a set of UI elements by traversing the DOM and keeping elements that are inside the viewport. For each retained element we record: (i) its bounding box in pixel coordinates, (ii) a coarse element type inferred from HTML/ARIA cues (used as a fallback label prior to refinement), and (iii) its textual content (from on-page text, with optional OCR fallback; §7.7.3). We also record auxiliary metadata such as the accessibility tree snapshot for potential downstream use.

**Hierarchy preservation.** A core goal of ScreenParse is to represent screens as structured states rather than flat sets of boxes. We therefore preserve *parent-child* relationships induced by the DOM: each element stores its parent index and a list of children indices (restricted to the extracted visible set). This yields a tree/forest structure that captures containment and grouping (e.g., a navigation bar containing tabs and buttons), which enables hierarchical serialization (ScreenTag; §7.7.5) and training objectives beyond single-element grounding.

**Multi-frame (iframe) handling.** Web pages often embed content in iframes. We extract elements from the main frame as well as embedded frames, and map their coordinates into a shared page coordinate system so that all boxes are comparable and can be jointly serialized.

### 7.7.3. TEXT EXTRACTION AND OCR

**Fine-grained text boxes.** In addition to per-element text, we optionally extract fine-grained text spans by collecting bounding rectangles of rendered text. This supports richer supervision and analysis (e.g., separating layout regions from text density).

**OCR fallback (optional).** When enabled and available, we run OCR (Tesseract) over element crops to fill missing or unreliable text for elements whose DOM text is empty (common for canvas-based or heavily scripted UIs). OCR is used as a fallback signal; DOM text remains primary when present.

### 7.7.4. FILTERING AND SAMPLE VALIDATION

Raw DOM extraction is intentionally dense and contains noise (layout wrappers, invisible artifacts, redundant overlapping boxes). We apply conservative filters designed to improve label quality while retaining semantically important UI.

**Geometric and visibility filtering.** We remove elements that are clearly unsuitable training targets, including: (i) invalid boxes (non-positive width/height), (ii) boxes that are almost entirely outside the viewport or have negligible visible overlap, (iii) *tiny* artifacts with area $< $ **4** pixels$^2$ (unless the element is recognized as an important interactive type), and (iv) overly large boxes that behave like page-wide wrappers (we set maximum area to **50%** of the viewport area during crawling, with a special-case exception for image regions).

**Duplicate suppression.** To reduce redundant annotations, we suppress near-duplicate boxes using IoU-based overlap checks with a threshold of **0.95**. When duplicates are detected, we preferentially keep boxes corresponding to interactive/semantically meaningful element types. To reduce redundancy at dataset level, we provide an hash based near-duplicate filter

with default Hamming radius **8**.

### 7.7.5. SCREENTAG SERIALIZATION

We represent each screen in a compact structure we called (**ScreenTag**) to serve as an efficient dense parsing target. Each element is serialized as a typed tag with discretized location tokens:

$$\texttt{<tag><x\_1><y\_1><x\_2><y\_2> text children </tag>},$$

where coordinates are given in left, top, right, bottom order and normalized to a **grid of 0-500**. We traverse the hierarchy depth-first and order siblings by top-left position to encourage stable reading order.

**Vocabulary and Tokenization.** To make structured generation efficient, we extend the tokenizer vocabulary with *single* special tokens for the ScreenTags, including opening/closing tags for the 55 UI classes and discretized location tokens for each coordinate bin on the **0–500** grid. This avoids producing tag strings as multi-token fragments, reduces the effective sequence length, and makes generation more efficient.

**Ground-truth cleanup.** Before writing labels, we apply an additional per-class duplicate cleanup with thresholds **IoU** $> 0.65$ and (for non-nestable classes such as Text/Button/Image) a containment-based duplicate rule with threshold **0.65**. For container-like classes we keep the largest box in a duplicate cluster; for atomic elements we keep the smallest.

The code for the Webshot pipeline will be released.

### 7.8. VLM-as-a-Judge Prompt for Annotation Quality Filtering

In the Webshot pipeline, we use qwen3-VL-8B-instruct as a VLM-as-a-judge to score screen annotations and filter low-quality pages in ScreenParse. Since the judge reports scores on a 0–100 scale, the threshold chosen is 70 to filter bad samples. The system and user prompts are provided below:

> **VLM-as-a-Judge Prompt (Qwen3-VL-8B-Instruct)**

```
_QUALITY_SYSTEM_PROMPT = '''You are a rigorous judge of UI bounding-box annotations.

You will be given:
- ONE image: a screenshot of a user interface with bounding boxes and text labels
    drawn on top of it. Each box corresponds to a single annotated UI element.
- The labels near the boxes show the element class (for example: Text, Button, Link,
    Image, Navigation Bar, etc.).

Your task is to evaluate **only the quality of the bounding box annotations**, NOT the
     correctness of the class names.

When judging, **treat the class label text as low priority**:
- Assume the class labels are mostly correct.
- Only use the labels to distinguish boxes or understand their intent.
- Do NOT heavily penalize minor class mistakes (e.g., Text vs Heading).
- Focus on: "Is there a sensible box here?" and "Are obvious UI elements missing or
    duplicated?"

-------------------------
EVALUATION DIMENSIONS
-------------------------

You must score the annotation quality using FOUR sub-scores and ONE overall score.

All scores are from 0 to 100, where:
- 100 = perfect in that dimension
- 80-99 = very good, only minor issues
- 60-79 = usable, but several noticeable issues
- 40-59 = poor, many issues
- 0-39 = extremely bad
```

1) COVERAGE / MISSING ELEMENTS (0-100)
   - Look for visually obvious, distinct UI elements that **should** be annotated:
     - buttons, main text blocks, headings, input fields, icons, major images, cards,
    menu items, etc.
   - Large and important items that should be covered:
     - main hero images, large central text, clearly clickable buttons or tabs,
    prominent fields.
   - Ignore tiny decorative details (small icons, background textures) unless they
    clearly look like clickable controls.
   - Don't forget that we are referring to the ui elements and text coverage, if there
     are empty areas, white/blank parts of the screen, it is correct that there shouldn
    't be boxes there.
   - High score (90-100): Almost every clear, medium-to-large UI element has a box.
   - Medium score (60-80): Some obvious UI elements are missing (e.g., a few main
    buttons or text blocks are unannotated).
   - Low score (<60): Many large, obvious UI elements have NO bounding boxes.

2) FALSE POSITIVES / SPURIOUS BOXES (0-100)
   - Penalize boxes that are not aligned with any visible UI element, such as:
     - Boxes in completely blank areas.
     - Boxes that repeat the same position but shifted somewhere else on the screen
    where nothing exists.
     - Boxes over pure background images or whitespace where there is no clear object
    or control.
   - Do NOT treat "container + child" as false positive if the container corresponds
    to a real region (e.g., window, navbar, card) and the child is a real button or
    text.
   - High score (90-100): Almost all boxes correspond to real UI elements or
    meaningful regions.
   - Medium score (60-80): Some boxes are obviously floating over empty space or
    nonsense regions.
   - Low score (<60): Many boxes are on empty background, duplicated far from any
    element, or clearly not aligned with any UI structure.

3) DUPLICATION / REDUNDANCY (0-100)
   Focus especially on SAME-CLASS duplications:

   - For NON-NESTABLE classes (for example: Text, Heading, Button, Checkbox, Radiobox,
     Switch, Slider, Text Input, Search Field, Image, Logo, Icon, etc.):
     - Two boxes of the same class that heavily overlap OR where one box is completely
    inside another usually indicate a problem.
     - Examples of bad cases:
       - A "Text" box entirely inside another "Text" box for the same text block.
       - A button annotated two or three times with slightly shifted boxes around the
    same visual button.
   - For container vs child classes (for example: Window + Button, Navigation Bar +
    Text, Card + Image):
     - It is OK for a container and its children to overlap (this is NOT a duplication
    ).
   - High score (90-100): Almost no obviously redundant same-class boxes; each UI
    element is annotated once.
   - Medium score (60-80): Some elements appear to have 2 overlapping same-class boxes.

   - Low score (<60): Many same-class boxes stacked or nested on top of each other in
    the same location, especially for Text and Button.

4) LOCALIZATION / ALIGNMENT (0-100)
   - Evaluate how well each bounding box fits its intended UI element.
   - Good annotation:
     - The box tightly covers the element, with small margins.
     - It does not cut off major parts of the element.
   - Penalize:

- Boxes that are much larger than the element and include large amounts of unrelated background.
- Boxes that cut off significant parts of the text/button/icon.
- High score (90-100): Boxes match element boundaries closely.
- Medium score (60-80): Some boxes are too loose or slightly misaligned, but the element is still clear.
- Low score (<60): Many boxes are badly misaligned, cutting off elements or covering very wrong region sizes.

--------------------------
OVERALL SCORE
--------------------------

Combine the four dimensions into a single **overall_quality** score (0-100).
Use approximately this weighting:
- Coverage / Missing: 40%
- False Positives:    25%
- Duplication:        20%
- Localization:       15%

Guidance:
- 90-100: Excellent annotation; safe to use as high-quality training data.
- 70-89: Good annotation; usable without manual review.
- 50-69: Borderline; has significant issues, might be filtered out if we want very clean data.
- 0-49: Bad annotation; should be excluded.

--------------------------
SPECIAL FAILURE PATTERNS TO PENALIZE HARD
--------------------------

Please penalize strongly when you see:

- Repeated "ghost" boxes:
  The same bounding box shape appears multiple times in unrelated places in the screenshot (especially when those copies are not aligned with any visible element). This is a strong indicator of broken coordinates.

- Repeated same-class boxes over one element:
  For example, several "Text" labels stacked over the same text string, or multiple overlapping "Button" boxes for one button.

- Missing entire regions:
  For example, a large hero image, main headline, or obvious call-to-action button with no annotation at all.

- Small irrelevant repeated boxes:
  Many tiny boxes of the same class (for example, small "Icon" or "Image" boxes) scattered around the screenshot that do not correspond to any real UI elements.

When such failure patterns are frequent across the image, the **overall_quality** should usually be below 40.

--------------------------
OUTPUT FORMAT
--------------------------

Return your answer as **pure JSON**, with no extra commentary.

Use this schema:

{

```
   "short_summary":         "<one-line summary of annotation quality, e.g. 'Good
     coverage but many duplicate Text boxes in irrelevant areas'>",
   "coverage_score":        <number 0-100>,
   "false_positive_score":  <number 0-100>,
   "duplication_score":     <number 0-100>,
   "localization_score":    <number 0-100>,
   "overall_quality":       <number 0-100>
}

Start with "short_summary" first - briefly describe the main quality issues (or lack
    thereof) before scoring.

Do not include any other keys or text outside this JSON.'''

_QUALITY_USER_PROMPT = "Please evaluate the annotation quality of this visualization
    image."
```

## 7.9. VLM Prompt for Class Refinement

We refine each element into the ScreenTag label set with a VLM that sees the entire page screenshot, the element crop, and a compact HTML/ARIA snippet. The following prompts map each element to a single class and an interactability flag.

**System prompt.**

**Class Refinement System Prompt**

```
You classify ONE UI element using:
 (1) the full-page screenshot,
 (2) a cropped image of the element,
 (3) a compact HTML-like snippet for the element.

Labeling rule:
- Choose exactly ONE label from the allowed list.
- Prefer the most specific, functionally correct label that best matches the element's
     purpose in its context.
- Favor function over appearance. If a more specific option exists, prefer it (e.g., "
    Search Field" over "Text Input" when it is clearly a search box; "Link" over "Text"
     if it navigates; "Button" over "Image" if it acts like a button).
- Use HTML/ARIA (role, type, href, aria-*, classes) to break ties. If still ambiguous,
     choose the closest single label from the list.

Interactability:
- Set interactable=true if a typical end-user can directly act on this element itself
    (click/tap/select/type/drag/scroll within it). Otherwise false (pure content/
    decoration or a passive container).

Output ONLY one JSON object on a single line:
{"label":"<one-of-allowed>","interactable":<true|false>}
```

**User prompt template.**

**Class Refinement User Prompt Template (55-class ScreenTag)**

```
Allowed labels:
- Table
- Column/Browser
- Button
- Utility Button
- App Icon
- Navigation Bar
```

- Status Bar
- Search Field
- Toolbar
- Tooltip
- Video
- Tab Bar
- Side Bar
- Slider
- Picker
- ContextMenu
- DockMenu
- EditMenu
- Image
- Scroll
- Switch
- File Icon
- Chart
- Window
- Screen
- List
- List Item
- PopUp Menu
- Steppers
- Toggles
- Text Input
- Rating Indicator
- Checkbox
- Radiobox
- Select
- Avatar
- Badge
- Alert
- Progress bar
- Bottom navigation
- Breadcrumb
- Page control
- Link
- Menu
- Pagination
- Tab
- Search Bar
- Date-Time picker
- Calendar
- Text
- Heading
- Code snippet
- Carousel
- Notification
- Logo

```
Element HTML snippet:
---------------------
<HTML_SNIPPET>
```

Instructions:
- Consider the crop FIRST to understand the element's visual identity.
- Use the full-page screenshot for surrounding context and function.
- Use HTML attributes (tag, role, type, aria-*, classes) as semantic hints.
- Output strictly one JSON object as specified (no extra text).

## 7.10. VLM Inference Prompts for Evaluation

For prompting-based baselines, we ask the model to extract all visible UI elements and return a JSON list with normalized bounding boxes, labels, and visible text. We use dataset-specific label sets: the 55-class ScreenTag taxonomy for ScreenParse (Tab. 9), the 8-class GroundCUA schema, and the 2-class ScreenSpot schema.

**Qwen3-VL and InternVL3 Inference Prompt**

```
You are a UI parser. Given a screenshot image, extract all visible UI elements.
Return JSON list with objects:
{"bbox_ltrb":[l,t,r,b], "label": "<type>", "text": "<visible text>"}.
The bbox_ltrb should be normalized to 0-1000. (l,t,r,b). Include all elements.

Allowed labels (ScreenParse / ScreenTag-55): [see Table 8]
Allowed labels (GroundCUA-8):
- Input Element
- Sidebar
- Information Display
- Button
- Navigation
- Visual Elements
- Menu
- Others
Allowed labels (ScreenSpot-2):
- icon
- text
```

