# OpenReview forum: "ScreenParse: Moving Beyond Sparse Grounding with Complete Screen Parsing Supervision"
_ICML.cc/2026/Conference — ICML 2026 regular_

### Official Review · Reviewer_YLod · 2026-03-10

**Soundness:** 3
**Presentation:** 2
**Significance:** 3
**Originality:** 3
**Overall Recommendation:** 5
**Confidence:** 3

**Summary:**

As per the 'ICML 2026 Call For Papers' in ICML official website, the authors are required to include 'Impact Statements'. However, this paper does not include such section, **raising concern of desk reject.**

This paper introduces ScreenParse, a large-scale dataset and a specialized model designed to solve the "screen parsing" in computer-use agents (CUA). While most existing datasets only label the specific button or icon needed for a single task (sparse grounding), ScreenParse provides dense, complete annotations for every visible element on a screen.

Overall, the study presents a central concept: moving from sparse, task-specific labels to holistic, "complete screen parsing" to provide agents with a better structural understanding of user interfaces. The authors discuss an important theme of efficiency, demonstrating that a compact, specialized model can outperform much larger general-purpose models when trained on high-quality, dense data.

**Compliance With Llm Reviewing Policy:**

Affirmed.

**Final Justification:**

The authors rebuttal solve my issue.

**Key Questions For Authors:**

N/A

**Limitations:**

Yes

**Strengths And Weaknesses:**

As per the 'ICML 2026 Call For Papers' in ICML official website, the authors are required to include 'Impact Statements'. However, this paper does not include such section, **raising concern of desk reject.**

**Strengths:**
- Massive Scale & Density: With 771K screenshots and 21M annotated elements across 55 categories, ScreenParse is significantly larger and more detailed than existing datasets like GroundCUA or ScreenSpot.
- Efficiency: The proposed ScreenVLM is remarkably compact (316M parameters). It achieves ~4x higher throughput and is ~6x smaller than larger-scale models while delivering superior parsing performance.
- Strong Generalization: The paper demonstrates that "dense supervision" provides structural priors that transfer well; finetuning larger foundation models on ScreenParse consistently improved their performance on out-of-distribution benchmarks.


**Major Weakness:**
- Domain Concentration: The dataset is predominantly web-centric. While the authors show some transfer to PC and Mobile, there is a noticeable performance drop (e.g., Recall@50 on Mobile is only 0.066), indicating a significant domain gap.
- Potential Annotation Noise: Despite VLM-based filtering, the "Webshot" pipeline relies on DOM data which is notoriously messy due to dynamic ads, overlays, and hidden elements, potentially leading to "noisy" ground truth in some samples.

**Minor Weakness:**
- The space utilization of figure can be improved. There are too many white space in figure 2, which is not professional.

---

> ### Author Rebuttal · Authors · 2026-03-30
>
> We thank the reviewer for the evaluation and the helpful feedback. We address each concern below.
>
> ## Impact Statement
>
> We thank the reviewer for pointing out this omission and will add the required Impact Statement to the revised paper. Dense screen parsing can benefit accessibility tools for visually impaired users, enable more reliable assistive technology and computer-use agents, and support automated UI testing workflows. On the other hand, improved GUI parsing could potentially be misused for unauthorized automated interaction with web services or UI-based social engineering. We mitigate these risks by constructing ScreenParse exclusively from publicly available URLs, releasing all resources under a research-only license with clear usage guidelines, and ensuring no personally identifiable information is included in the dataset. We apologize for the oversight in the initial submission.
>
> ## W1 (Major): Domain Concentration
>
>  We acknowledge ScreenParse's web-centric nature and discuss this explicitly in our Limitations section (Sec. 6). However, the cross-domain transfer results are stronger than they may initially appear. The low Mobile Recall@50 of 0.066 is specific to ScreenVLM's zero-shot transfer on unseen widget styles, the value of ScreenParse as dense supervision generalizes more broadly. Fine-tuning other models on ScreenParse yields consistent improvements across non-web domains: on GroundCUA (87 real desktop applications), InternVL3-2B improves from 0.025 to 0.203 PageIoU; on ScreenSpot, OmniParser improves on both PC (Recall 0.483→0.536) and Mobile (0.489→0.552). These cross-model, cross-domain gains confirm that web-based dense supervision provides transferable structural priors even for UI distributions not seen during training. Extending ScreenParse to native desktop and mobile UIs is our most immediate next step to close the remaining precision gap.
>
> ## W2 (Major): Potential Annotation Noise
>
>  We appreciate this concern; DOM-based extraction is indeed inherently noisy, which is precisely why we designed a multi-stage quality pipeline rather than relying on raw DOM output. Our pipeline applies four successive filtering stages: (1) rule-based geometric and visibility filtering that removes invalid boxes, off-screen elements, tiny artifacts (<4 px²), oversized DOM wrappers (>50% viewport), and near-duplicate boxes (IoU >0.95); (2) VLM-based label refinement, where Qwen3-VL-8B reclassifies each element using the full-page screenshot, element crop, and HTML/ARIA snippet jointly; (3) VLM-as-a-judge scoring across four quality dimensions (coverage, false positives, duplication, localization), discarding pages below a 0.70 threshold; and (4) dataset-level near-duplicate removal via perceptual hashing (Hamming distance <8).
>
> To directly quantify the resulting annotation quality, we conducted a human audit for this rebuttal on 100 randomly sampled pages (3,173 UI elements). The results indicate high label quality: 97.73% element-level micro accuracy, 96.77% mean per-page accuracy, and Micro/Macro F1 of 0.989/0.921, with 94/100 pages achieving ≥90% accuracy. We will include these results along with per-class precision/recall in the revised paper.
>
> ## Minor: Figure 2 White Space
> We agree and will improve the space utilization and text readability of Figure 2 in the revised paper.

---

> > ### Author Rebuttal · Reviewer_YLod · 2026-04-02
> >
> > Thank the author for their rebuttal. I'd like to raise the score to 5.
> >
> > However, the author should carefully revise the figures in the paper, including figure 2 and figure 4.

---

> > > ### Author Response · Authors · 2026-04-05
> > >
> > > Thank you for the update and for raising your score.
> > >
> > > We will revise Figures 2 and 4 for better space utilization and readability, and will include the required Impact Statement in the revised paper.
> > >
> > > We appreciate the constructive feedback.

---

### Official Review · Reviewer_xxiT · 2026-03-11

**Soundness:** 3
**Presentation:** 3
**Significance:** 3
**Originality:** 3
**Overall Recommendation:** 5
**Confidence:** 4

**Summary:**

This paper introduces ScreenParse, a large-scale densely-annotated dataset for complete UI screen parsing (771K screenshots, 21M elements, 55 classes), generated via an automated Webshot pipeline. The authors train ScreenVLM, a compact 316M-parameter vision-language model that outputs structured ScreenTag sequences, and show that it outperforms much larger prompted VLMs on dense parsing. They further demonstrate that fine-tuning existing foundation VLMs and detectors on ScreenParse consistently improves performance on both in-domain and out-of-distribution benchmarks.

**Compliance With Llm Reviewing Policy:**

Affirmed.

**Final Justification:**

The authors have addressed my concerns, so I have raised score to 5.

**Key Questions For Authors:**

Please refer to the weaknesses mentioned above.

**Limitations:**

Yes

**Strengths And Weaknesses:**

# Strengths
S1: **Well-motivated problem with practical significance.** The paper convincingly argues that sparse, action-oriented supervision is insufficient for building robust screen perception. The distinction between single-target grounding and complete screen parsing is clearly articulated.

S2: **Strong dataset contribution at scale.** ScreenParse is larger and more fine-grained than prior complete-annotation datasets, though it might be in a different domain. The automated Webshot pipeline is well-engineered, with multiple quality control stages. The prompts for these stages are fully provided in the appendix, which is commendable for reproducibility.

S3: **Consistent and transferable gains across model families.** One of the paper's most convincing findings is that ScreenParse supervision is not model-specific. This strongly supports the claim that dense screen-level supervision provides transferable structural priors.


# Weaknesses

W1: **Web-only domain limits generalization claims.** ScreenParse is constructed entirely from web pages, yet the paper makes broad claims about "holistic screen understanding" and "transferable structural priors." The GroundCUA and ScreenSpot transfers partially validate this, but the results also expose clear domain gaps: on ScreenSpot Mobile, ScreenVLM's Recall@50 drops to 0.066, and on PC it reaches only 0.222. The paper acknowledges this in limitations but the framing in the introduction and abstract could be more tempered. The claim that dense supervision provides transferable priors should be qualified more carefully, given that transfer to non-web domains shows mixed results.

W2: **Label PageIoU scores remain quite low across the board.** Even ScreenVLM achieves only 0.197 Label PageIoU on ScreenParse and 0.043 on GroundCUA. This suggests that while spatial coverage (PageIoU) is reasonable, the model struggles significantly with accurate element typing. The paper does not provide a detailed error analysis or per-class breakdown showing which of the 55 classes are well-predicted versus problematic. Understanding whether the issue is class confusion among similar types (e.g., Button vs. Utility Button, Text vs. Heading) or systematic failures on rare classes would substantially strengthen the analysis.

W3: **No downstream task evaluation.** The paper motivates complete screen parsing as a prerequisite for effective agent planning and action execution, but no experiment evaluates whether ScreenVLM's dense parsing actually improves downstream agent performance on grounding (e.g. UI-Vision, ScreenSpot-Pro) and interactive benchmarks (e.g., WebArena, OSWorld). The gap between "better PageIoU" and "better agent task success" is non-trivial and unvalidated. This weakens the significance claim.

W4: **Annotation quality assessment is limited.** While the VLM-as-a-judge filter is a reasonable automated approach, the paper provides only a brief mention of "targeted human validation on held-out samples to calibrate thresholds". More details on this will consolidate the work even more.

---

> ### Author Rebuttal · Authors · 2026-03-31
>
> We thank the reviewer for the thorough review, recognition and address each point below.
>
> ## W1: Web-Only Domain and Generalization Claims
> We agree with the reviewer and will temper the framing in the abstract and introduction accordingly. That said, transfer evidence is stronger than Recall@50 alone suggests. On ScreenSpot, ScreenVLM's PixCov reaches 0.839 (PC) and 0.847 (Mobile), covering the vast majority of annotated target pixels on unseen UI distributions, the low Recall@50 reflects difficulty producing tight, element-level boxes for unfamiliar widget styles, not a failure to recognize UI structure. On GroundCUA (87 platforms, including native desktop applications), ScreenVLM achieves PageIoU 0.251 vs. 0.060 for Qwen3-VL-8B, a 4× improvement despite the domain shift. The benefit is model-agnostic: fine-tuning InternVL3-2B on ScreenParse improves GroundCUA PageIoU from 0.025 to 0.203 and ScreenSpot PC PixCov from 0.109 to 0.628.
> In the revision, we will distinguish strong transfer of structural priors (high PixCov, consistent gains across model families) from remaining element-level precision gaps on non-web UIs, framing the "transferable priors" claim with appropriate nuance.
>
> ## W2: Low Label PageIoU Scores
> Two factors explain the lower Label PageIoU:
> (1) Fine-grained taxonomy: ScreenParse's 55 classes include visually similar pairs (e.g., Button vs. Utility Button, Text vs. Heading vs. Link). Exact label matching penalizes spatially correct predictions assigned a neighboring class.
> (2) Cross-dataset label mismatch: GroundCUA uses 8 coarse classes, creating systematic ambiguity in the many-to-few mapping (e.g., "Navigation Bar," "Tab Bar," "Side Bar," and "Breadcrumb" all collapse to "Navigation"). Additionally, ~40% of GroundCUA GT elements lack type annotations, further depressing label-sensitive metrics.
> Per-class confusion analysis on one-to-one matches after successful localization (rates per 1,000 GT elements, aggregated over 4 models):
>
> **ScreenParse - top confusion pairs:**
>
> | Confusion Pair | Rate / 1K |
> |---|---|
> | Utility Button → Button | 116 |
> | Button → Link | 67 |
> | Text → Link | 52 |
> | Link → Text | 35 |
> | Button → Utility Button | 34 |
> | Link → Button | 25 |
> | Logo → Image | 20 |
>
> **GroundCUA - top confusion pairs:**
>
> | Confusion Pair | Rate / 1K |
> |-|-|
> | Menu → Navigation | 598 |
> | Input Element → Info Display | 300 |
> | Navigation → Info Display | 299 |
> | Navigation → Button | 250 |
> | Input Element → Button | 225 |
> | Menu → Info Display | 200 |
>
> Dominant errors concentrate in semantically adjacent classes. On ScreenParse, confusion occurs mainly within text/link/button boundaries. On GroundCUA, errors are dominated by overlap among broad coarse categories. PageIoU remains the more informative cross-dataset metric, as it measures structural coverage independently of taxonomy alignment.
>
> ## W3: No Downstream Task Evaluation
> We agree that downstream evidence strengthens the paper's motivation. The paper's core claim is that dense supervision provides stronger structural priors than sparse supervision; downstream agent pipelines involve many additional factors (action space, reasoning, prompting) beyond perception. That said, we have conducted controlled experiments for this rebuttal.
> Using an OmniParser-v2-style Set-of-Mark pipeline with Qwen3-VL-8B-Instruct, we swap only the detector (OmniParser v2 vs. ScreenParse-trained YOLO), holding all other components fixed. ScreenParse-trained perception improves the primary metric across all five benchmark settings: ScreenSpot (+2.3 pts Action Acc.), ScreenSpot-Pro (+2.0 pts), and all three Mind2Web splits (up to +3.0 pts Step Success). Per-domain ScreenSpot breakdowns confirm gains across web, mobile, and desktop. Full tables are in our response to **Reviewer t63k (W2 & Q4).**
> These preliminary results directly address the concern: stronger ScreenParse-trained perception yields consistent downstream GUI-agent gains across multiple benchmarks and domains. We will include these results in the revision.
>
> ## W4: Annotation Quality Assessment
> We agree that more detail on human validation would strengthen confidence in the dataset. We have conducted a quantitative human audit on the final filtered dataset: annotators independently reviewed 100 randomly sampled pages (3,173 elements), classifying each bounding box against our 55-class taxonomy. Results indicate high annotation quality:
> Element-level micro accuracy: 97.73% (3,101/3,173 correct)
> Mean per-page accuracy: 96.77%
> Micro F1: 0.9888 / Macro F1: 0.9208
> 84/100 pages achieved ≥95% accuracy; 94/100 pages achieved ≥90%
> This audit covers class label correctness for detected elements. Coverage and localization quality are separately validated by the VLM-as-a-judge filter, which scores each page on coverage, false positives, duplication, and localization before acceptance. The full per-class precision/recall breakdown and evaluation tool will be included in the supplementary material.

---

> > ### Author Rebuttal · Reviewer_xxiT · 2026-04-03
> >
> > Thank the authors for the detailed reply. I have no more questions for W2 and W4 (but please add the content to the next revision of the paper). For W3, I went through the authors' reply to Reviewer t63k (W2 & Q4), and I want to ask why the authors say they can not evaluate OSWorld-G while they actually evaluated on ScreenSpot and ScreenSpot-pro. Correct me if I am wrong, but I think they are all GUI grounding benchmarks with a similar task style. If this is actually feasible, combined with my comment on W1, it would be good to see if the proposed method can perform well on Desktop environment (like UI-Vision and OSWorld-G), where the parsing tasks are harder and more useful.

---

> > > ### Author Response · Authors · 2026-04-05
> > >
> > > Thank you for the follow-up. We appreciate the reviewer raising this point. The reviewer is right that OSWorld-G is a grounding benchmark feasible under our existing setup, and our earlier framing was imprecise. At the time of the initial rebuttal, we had not yet completed the evaluation under our controlled protocol, but we have now done so, for OSWorld-G, during the discussion period.
> > >
> > > We keep the full OmniParser-v2-style pipeline fixed (OCR, icon captioning, SoM rendering, prompting, `Qwen3-VL-8B-Instruct` backend) and swap only the detector (`OmniParser v2` vs. `ScreenParse`-trained YOLO):
> > >
> > > | Detector | Text Matching | Element Recognition | Layout Understanding | Fine-grained Manipulation | Overall |
> > > |---|---:|---:|---:|---:|---:|
> > > | ScreenParse (YOLO) | 61.69 | 50.91 | 56.13 | 29.53 | 48.82 |
> > > | OmniParser | 59.00 | 50.30 | 53.36 | 25.50 | 46.67 |
> > > | Δ | +2.68 | +0.61 | +2.77 | +4.03 | +2.16 |
> > >
> > > These preliminary results provide additional evidence on a harder desktop-oriented benchmark beyond the results already reported in our rebuttal: ScreenParse improves all four grounding-relevant capability categories on this harder desktop-oriented benchmark, with the largest gain on fine-grained manipulation (+4.03). Combined with the positive results on ScreenSpot-Pro, the ScreenSpot desktop split, and GroundCUA (87 desktop applications), this strengthens the evidence that structural priors learned from ScreenParse transfer beyond the web domain.
> > >
> > > So, more precisely, our original point was not that OSWorld-G is infeasible, but that we had not yet completed it under the same validated detector-swap protocol at the time of the first rebuttal. We appreciate the reviewer pushing us on this point and will include these results in the revised paper.

---

### Official Review · Reviewer_rXYu · 2026-03-12

**Soundness:** 3
**Presentation:** 2
**Significance:** 4
**Originality:** 3
**Overall Recommendation:** 5
**Confidence:** 5

**Summary:**

The paper addresses the limitations of sparse supervision in vision-language models for computer-use agents (CUAs) by advocating for complete screen parsing. To achieve this, the authors introduce Webshot, an automated data collection pipeline that extracts DOM-aligned UI elements to create ScreenParse, a large-scale dataset featuring 771K web screenshots with over 21 million dense, hierarchically structured annotations across 55 UI categories. Leveraging this dataset, they propose ScreenVLM, an ultra-compact (316M-parameter) and efficient vision-language model trained with a novel structure-aware weighted cross-entropy loss. The authors demonstrate that ScreenVLM outperforms much larger foundation models on dense parsing tasks and generalizes well to out-of-distribution desktop and mobile interfaces, while also showing that the ScreenParse dataset can significantly boost the performance of existing foundation and detector-based models.

**Compliance With Llm Reviewing Policy:**

Affirmed.

**Final Justification:**

The authors successfully addressed my primary concerns during the rebuttal, specifically by providing the missing end-to-end downstream agent evaluations. Given the clear utility of the dataset to the community and the newly validated impact on downstream tasks, I confidently maintain my recommendation to accept.

**Key Questions For Authors:**

1. End-to-End Agentic Evaluation: The paper heavily motivates complete screen parsing as a prerequisite for downstream computer-use agents (CUAs) to plan and execute actions. However, the evaluation is strictly perception-based (e.g., IoU, Recall). Do you have any preliminary results showing how ScreenVLM, or models fine-tuned on ScreenParse, perform on end-to-end agentic task benchmarks (such as OSWorld, Mind2Web, or WebArena)?

2. End-to-End VLM vs. Detector + LLM Pipelines: Table 3 shows that detector models (e.g., OmniParser, RT-DETRv2) achieve significantly higher exact localization scores (mAP@50) than ScreenVLM on out-of-distribution screens. You note that detectors lack "language-grounded" states, but standard practical pipelines simply pair a detector with an LLM to achieve this. What is the concrete advantage of using the end-to-end ScreenVLM over a state-of-the-art "Detector + LLM" pipeline for GUI agents?

3. Spatial and Structural Biases in Data Collection: Appendix 7.6 reveals that Webshot only captures the "top-of-page viewport without scrolling" and hard-filters container elements exceeding 50% of the viewport area. How do these constraints affect the model's ability to generalize to lower-page elements (e.g., footers, bottom-anchored forms) or modern web applications that utilize massive split-screen layouts/overlays?

4. Architectural and Clarification Details: There are a few technical ambiguities that hinder full reproducibility:
(a) What specific pooling mechanism and reduction factor are utilized in the "Projection + Pooling" step referenced in Figure 4?
(b) Does quantizing coordinates to a 500-bin grid hurt the precise localization of tightly packed UI elements on high-resolution screens (e.g., 1440x900)?
(c) Could you clarify the VLM-as-a-judge filtering threshold (Figure 2 illustrates a score of "90" for acceptance, while Appendix 7.7 states "0.70")?

**Limitations:**

Yes. The authors transparently acknowledge the primary limitation of their work, specifically the domain gap between web-based training data and native environments.

For the camera-ready version, the authors might consider briefly noting a few minor technical constraints of the data collection pipeline as well. For instance, relying purely on the DOM may miss elements in canvas-based web apps, and capturing only the top-of-page viewport introduces a slight spatial bias. Additionally, a brief sentence acknowledging the general dual-use nature of GUI parsing models (e.g., potential misuse for aggressive automated scraping) would round out the broader impact statement. However, these are minor points and do not detract from the overall quality and utility of the submission.

**Strengths And Weaknesses:**

Strengths

Dataset as a Rising Tide (Significance): A major strength of this work is that the ScreenParse dataset provides value beyond the authors' own model. The experiments clearly demonstrate that fine-tuning other foundation VLMs (like InternVL3 and Qwen3-VL) and detector-based models (like OmniParser) on this dense supervision significantly boosts their performance.

Efficiency and Practical Utility (Significance): ScreenVLM successfully proves that an ultra-compact (316M parameters) architecture can outperform massive foundation models on specific parsing tasks. The inclusion of latency and throughput metrics highlights its practical viability for real-time, on-device agent applications.

Structure-Aware Loss (Originality/Soundness): The novel loss function is well-justified. The ablation studies effectively prove that upweighting structural tokens (tags and coordinates) over text tokens prevents the model from over-optimizing for text transcription, leading to better out-of-distribution generalization.

----------------------

Weaknesses

Unsubstantiated "Agent" Claims (Soundness): The paper repeatedly motivates the need for complete screen parsing by claiming it is essential for downstream computer-use agents (CUAs) to plan and act. However, the authors only evaluate perception metrics (IoU, Recall). To validate this core claim, the model must be evaluated on an end-to-end agentic benchmark (such as WebArena, Mind2Web, or OSWorld) to prove that better dense parsing actually translates to higher task success rates.

Poor Localization on Native UIs (Soundness): When transferring to out-of-distribution environments like the PC and Mobile splits of ScreenSpot, ScreenVLM experiences a severe drop in Recall@50. While the high PixCov scores indicate the model roughly knows where elements are, it fundamentally fails to draw precise, accurate bounding boundaries on native UIs, limiting its current utility as a general OS agent.

Incomplete Baseline Comparisons (Soundness): Detector models (e.g., OmniParser, RT-DETRv2) significantly outperform ScreenVLM on exact localization metrics like mAP@50. The authors dismiss detectors by arguing they do not produce "language-grounded screen states." However, in practice, standard GUI agent pipelines pair a detector with an LLM to achieve exactly this. The authors should justify why an end-to-end VLM approach is superior to a mature "Detector + LLM" pipeline.

Brittle DOM-Extraction Reliance (Soundness/Methodology): The Webshot pipeline relies heavily on the HTML DOM to extract elements. The authors do not adequately address how this pipeline handles modern canvas-based web applications (like Google Docs or Figma), where the visual UI is entirely decoupled from the DOM structure.

Spatial and Structural Biases in Data Collection (Significance): As noted in the Appendix, the Webshot pipeline only captures the "top-of-page viewport without scrolling." This introduces a severe spatial bias, effectively teaching the model that lower-page elements (like footers or bottom-anchored submit buttons) do not exist. Furthermore, hard-filtering container elements that take up more than 50% of the screen contradicts the stated goal of capturing true hierarchical structures, as many modern web layouts utilize massive split-screen containers or full-page overlays.

Missing Critical Related Work (Originality): The paper fails to discuss two highly relevant, state-of-the-art lineages in GUI grounding. First, CogAgent (CVPR 2024), an 18B VLM that introduced a dual-resolution architecture specifically designed to solve the exact dense UI parsing and tiny-text recognition problems discussed in this paper. Second, the Ferret-UI family (Apple, 2024-2026), particularly the recent Ferret-UI Lite, which targets the exact same niche as ScreenVLM: building highly compact (~3B parameter), on-device GUI agents.

Presentation and Clarity Issues (Presentation): Figure 2 appears rushed; the text inside the pipeline boxes is illegibly small compared to the manuscript font, and there is excessive wasted whitespace. Additionally, there is a contradiction regarding the VLM-as-a-judge filtering mechanism: Figure 2 illustrates a score of "90" for acceptance, whereas Appendix 7.7 explicitly states the threshold is "0.70".

---

> ### Author Rebuttal · Authors · 2026-03-31
>
> We thank the reviewer and address each point below.
> ## W1 & Q1: End-to-End Agentic Evaluation
>
> We agree that downstream evidence is important for the agentic motivation, and in response to this concern we conducted additional controlled experiments to test whether improved screen parsing transfers to downstream GUI-agent performance.
>
> We use the OmniParser-v2-style Set-of-Mark pipeline with Qwen3-VL-8B-Instruct, swapping only the detector (OmniParser v2 vs. ScreenParse-trained YOLO). All other components (OCR, icon captioning, SoM rendering, prompting, VLM backend) are held fixed, isolating the perception module. Under this controlled setup, ScreenParse-trained perception improves the primary downstream metric on all five benchmark settings (ScreenSpot, ScreenSpot-Pro, and three Mind2Web splits), with supporting metrics and per-domain breakdowns confirming gains across web, mobile, and desktop. Please refer to the detailed tables in our **response to Reviewer t63k (W2 & Q4)**. These preliminary results directly address the core concern: stronger perception yields better downstream performance across multiple benchmarks and domains.
>
> ## W2: Poor Localization on Native UIs
>
> We acknowledge the Recall@50 drop on PC and Mobile. Fine-tuning other models on ScreenParse yields clear non-web improvements: OmniParser on ScreenSpot PC (Recall 0.483→0.536) and Mobile (0.489→0.552); InternVL3-2B on GroundCUA PageIoU 0.025→0.203. These cross-model gains confirm transferable structural priors. Extending annotation to native UIs is our immediate next step.
>
> ## W3 & Q2: End-to-End VLM vs. Detector+LLM
>
> An end-to-end VLM enables fine-tuning with shared gradients across perception and language, whereas a Detector+LLM pipeline trains modules independently, making joint optimization less straightforward. Importantly, ScreenParse benefits both: Our experiments show gains for detectors and VLMs alike, so the dataset contribution is agnostic to the architectural choice.
>
> ## W4: Brittle DOM-Extraction (Canvas-Based Apps)
>
> Our VLM-as-a-judge filter (Sec. 3.2, Appendix 7.7) addresses this: pages where DOM-extracted boxes do not align with the visible UI receive low quality scores and are discarded. We also support OCR fallback for text extraction in such cases (Appendix 7.6).
>
> ## W5 & Q3: Spatial and Structural Biases
>
> (1) The Webshot pipeline supports both viewport-only and full-page capture. We chose viewport-only as full-page rendering introduces noise from scroll-triggered state changes and bounding box misalignment. This underrepresents lower-page elements, but does not prevent learning generalizable priors: ScreenVLM transfers well to GroundCUA (87 platforms) and ScreenSpot. We will analyze full-page captures in the revision. (2) The 50% viewport area filter targets semantically empty DOM wrappers, not content-bearing containers like navbars or modals which fall well below this threshold. We will quantify the fraction of semantically meaningful filtered elements in the revision.
>
> ## W6: Missing Related Work
>
> We thank the reviewer for highlighting these relevant works. CogAgent (Hong et al., CVPR 2024) uses a dual-resolution 18B VLM with 400K web screenshots for element-level QA. Ferret-UI and successors target mobile/multi-platform UI at 7–13B (~3B for Lite) with 13 classes. These are valuable contributions, and we will incorporate them in the revised related work. Our work differs in: (1) ScreenParse provides full screen-level annotations instead of element-level QA pairs. (2) a 55-class schema based on Apple HIG, Material UI, and Fluent UI vs. 13 coarse classes; (3) hierarchical container structure via ScreenTag; (4) our multi-stage Webshot pipeline (heuristic filtering → VLM refinement → VLM-as-judge) goes beyond basic HTML cleaning or detector-based annotation.
>
> ## Q4: Architectural Details
>
> (a) The connector follows the Idefics3 architecture: pixel-shuffle (scale_factor=4) reduces 1,024 SigLIP2 tokens to 64 by grouping 4×4 adjacent patches and concatenating features (768→12,288), followed by a bias-free linear projection (12,288→576) into Granite-165M. The pixel shuffle is lossless, spatial information is rearranged, not discarded. We will add these details for reproducibility. (b) At 1440×900, 500 bins yield ~2.88×1.80 px/bin, sufficient for most UI elements. Our Recall@50 and mAP@50 results (Tab. 3) confirm quantization is not a bottleneck. Finer grids for ultra-dense screens are a natural extension. (c) Addressed in W7 below.
>
> ## W7: Presentation Issues
> (1) We will redesign Figure 2 with larger text and tighter layout. (2) The "90" in Figure 2 is an illustrative example score (which passes the threshold), not the threshold itself. The actual threshold of 0.70 is in Appendix 7.7. We will clarify in the caption.
>
> ## Broader Impact
>
> We will add dual-use considerations: accessibility benefits alongside risks such as unauthorized scraping or UI automation misuse.

---

> > ### Author Rebuttal · Reviewer_rXYu · 2026-04-04
> >
> > I thank the authors for the detailed and constructive rebuttal. Providing the new downstream Set-of-Mark experiments on ScreenSpot and Mind2Web effectively resolves my primary concern regarding end-to-end agentic evaluation. The clarifications on the architectural details (pooling and quantization), along with the commitment to update Figure 2, the related work, and the impact statement, adequately address my remaining points.
> >
> > My concerns have been fully resolved, and I will maintain my score of Accept.

---

> > > ### Author Response · Authors · 2026-04-05
> > >
> > > Thank you for the constructive discussion and for indicating that your concerns were resolved. For completeness, during the discussion period we also completed one additional controlled OSWorld-G evaluation under the same detector-swap setup used in our rebuttal experiments:
> > >
> > > | Detector | Text Matching | Element Recognition | Layout Understanding | Fine-grained Manipulation | Overall |
> > > | --- | ---: | ---: | ---: | ---: | ---: |
> > > | ScreenParse (YOLO) | 61.69 | 50.91 | 56.13 | 29.53 | 48.82 |
> > > | OmniParser | 59.00 | 50.30 | 53.36 | 25.50 | 46.67 |
> > > | Δ | +2.68 | +0.61 | +2.77 | +4.03 | +2.16 |
> > >
> > > We include this result as an additional data point because your review emphasized the need for downstream / agentic evidence rather than perception-only gains. It is consistent with the same conclusion from our rebuttal experiments: under a controlled screenshot-agent pipeline, improving the perception module also improves downstream grounding performance. We will include this result in the revised paper.

---

### Official Review · Reviewer_t63k · 2026-03-14

**Soundness:** 3
**Presentation:** 3
**Significance:** 3
**Originality:** 2
**Overall Recommendation:** 4
**Confidence:** 4

**Summary:**

This paper studies complete screen parsing for GUI perception and argues that existing datasets for GUI agents mainly rely on sparse grounding annotations that fail to capture the full structural state of a screen. To address this limitation, the authors introduce ScreenParse, a large-scale dataset containing 771k web screenshots and 21M UI element annotations across 55 semantic classes, generated through an automated Webshot pipeline combining DOM extraction, heuristic filtering, and VLM-based refinement. Using this dataset, the authors train ScreenVLM, a compact 316M vision–language model that outputs a structured ScreenTag representation with a structure-aware loss emphasizing layout and type tokens. Experiments show that ScreenVLM outperforms larger prompted VLMs on dense screen parsing metrics and transfers reasonably well to public GUI grounding benchmarks.

**Compliance With Llm Reviewing Policy:**

Affirmed.

**Final Justification:**

The paper presents a large-scale and well-constructed dataset with clear empirical improvements across multiple benchmarks. The dense annotation pipeline and overall engineering effort are valuable, and the dataset is likely to serve as a useful resource for future research on GUI understanding and multimodal agents.

Overall, I would like to thank the authors for the high-quality rebuttal and maintain my positive evaluation.

**Key Questions For Authors:**

1. Annotation Quality:
   Can the authors provide a quantitative evaluation of annotation quality (e.g., human audit, per-class precision/recall, or inter-annotator agreement)? This would significantly strengthen confidence in the dataset.
2. Dataset Split Hygiene:
   How do the authors ensure that train/validation/test splits avoid domain or URL overlap? Clarifying the deduplication procedure would improve experimental rigor.
3. Loss Weight Sensitivity:
   Did the authors explore different values for the structure-aware loss weights? A sensitivity analysis would clarify whether the observed improvements are robust.
4. Generalization to broader grounding benchmarks:
   Can the authors evaluate the model on additional GUI grounding benchmarks (e.g., OSWorld-G, MMBench-GUI...) to better assess generalization across different CUA settings?

**Limitations:**

Yes

**Strengths And Weaknesses:**

## Strengths

1. Large-scale dense dataset and scalable data pipeline.
   A major strength of the paper is the introduction of ScreenParse, a large-scale dataset with 771k screenshots and over 21M UI element annotations across 55 semantic classes. Compared with prior GUI grounding datasets that provide sparse annotations, ScreenParse offers dense, screen-level supervision covering nearly all visible elements. The proposed Webshot pipeline provides a scalable way to automatically collect such annotations through DOM extraction, heuristic filtering, and VLM-based refinement, significantly reducing the cost of manual labeling and enabling dataset construction at web scale.

2. Practical model design with strong empirical performance and efficiency.
   The proposed ScreenVLM model provides a lightweight yet effective approach for dense screen parsing. By using a compact 316M vision–language model that generates structured ScreenTag representations, the system achieves strong performance on dense parsing metrics while maintaining high efficiency. Experimental results across multiple datasets (ScreenParse, GroundCUA, and ScreenSpot) show consistent improvements from dense supervision, and the model demonstrates clear latency and throughput advantages compared with larger foundation VLMs.

3. Clear problem framing and well-presented methodology.
   The paper clearly motivates the shift from sparse grounding to holistic screen parsing, arguing that full structural understanding is essential for robust computer-use agents. The overall presentation is well organized, with clear separation between dataset construction, model design, and experiments. Figures illustrating the Webshot pipeline and ScreenTag representation help explain the approach, and evaluation metrics such as PageIoU, Recall@50, and PixCov are clearly defined, making the methodology easy to follow and reproduce.

## Weaknesses

1. Limited validation of dataset quality.
   A key concern is the reliability of the automatically generated annotations. The ScreenParse dataset is produced through DOM extraction and VLM-based filtering, but the paper provides limited quantitative validation of annotation quality. In particular, there is little human evaluation or detailed error analysis (e.g., per-class precision/recall or annotation noise statistics). Without stronger verification, it is difficult to fully assess the accuracy of the dense annotations and their potential impact on training and evaluation.

2. Missing experiments and evaluation breadth.
   The empirical evaluation could be further strengthened. While the paper focuses on screen parsing and grounding benchmarks, evaluating on additional CUA-related grounding benchmarks (e.g., OSWorld-G or MMBench-GUI) would provide a more comprehensive assessment of generalization in realistic agent settings. Moreover, the study mainly reports parsing metrics and does not analyze how models trained on ScreenParse benefit downstream GUI agent tasks. Experiments that train or evaluate GUI agents using ScreenParse for training would better demonstrate its practical impact on agentic capability.

3. Limited novelty and potential generalization constraints.
   While the dataset contribution is valuable, the modeling component shows relatively limited architectural novelty. ScreenVLM largely adapts existing document-to-markup VLM architectures with modest modifications. Furthermore, the dataset is primarily web-based, which may limit generalization to desktop or mobile UI environments. The results also indicate that detector-based parsers still outperform the proposed approach on certain localization metrics, suggesting that the practical advantages over specialized detection pipelines remain somewhat limited.

---

> ### Author Rebuttal · Authors · 2026-03-31
>
> We thank the reviewer for the thoughtful evaluation and recognition of our contributions.
>
> ## W1 & Q1 Annotation Quality Validation
>
> We conducted a human audit on 100 randomly sampled pages (3,173 elements), classifying each bounding box against our 55-class taxonomy. Results: 97.73% element-level micro accuracy, 96.77% mean per-page accuracy, Micro/Macro F1 of 0.9888/0.9208, with 94/100 pages achieving ≥90% accuracy. See **response to Reviewer xxiT (W4)** for details. We will include per-class precision/recall in the revised paper.
>
> ## W2 & Q4: Evaluation Breadth and Broader Benchmarks
>
> The suggested benchmarks (e.g., OSWorld-G, MMBench-GUI) require a full agent pipeline, beyond this paper's perception focus. That said, we conducted controlled downstream experiments for this rebuttal.
>
> We use the OmniParser-v2-style Set-of-Mark pipeline with Qwen3-VL-8B-Instruct, swapping only the detector (OmniParser v2 vs. ScreenParse-trained YOLO). All other components are held fixed, isolating the perception module. ScreenParse improves the primary metric on all five settings:
>
> | Benchmark | Metric | OmniParser v2 | ScreenParse (YOLO) | Δ |
> |---|---|---|---|---|
> | ScreenSpot | Action Acc. | 77.8% | 80.1% | +2.3 |
> | ScreenSpot-Pro | Action Acc. | 28.5% | 30.5% | +2.0 |
> | Mind2Web (website) | Step Success | 24.1% | 27.1% | +3.0 |
> | Mind2Web (task) | Step Success | 26.8% | 27.0% | +0.2 |
> | Mind2Web (domain) | Step Success | 28.7% | 29.6% | +0.9 |
>
> Supporting metrics:
>
> | Benchmark | Metric | OmniParser v2 | ScreenParse (YOLO) | Δ |
> |---|---|---|---|---|
> | ScreenSpot | Text Acc. | 83.8% | 87.1% | +3.3 |
> | ScreenSpot | Icon Acc. | 70.6% | 71.1% | +0.5 |
> | Mind2Web (website) | Element Acc. | 31.1% | 33.9% | +2.9 |
> | Mind2Web (website) | Action F1 | 80.7% | 81.1% | +0.4 |
> | Mind2Web (task) | Element Acc. | 31.6% | 32.3% | +0.7 |
> | Mind2Web (task) | Action F1 | 84.2% | 84.4% | +0.2 |
> | Mind2Web (domain) | Element Acc. | 34.2% | 34.6% | +0.4 |
> | Mind2Web (domain) | Action F1 | 83.5% | 83.7% | +0.2 |
>
> ScreenSpot domain breakdown:
>
> | ScreenSpot Domain | OmniParser v2 | ScreenParse (YOLO) | Δ |
> |---|---|---|---|
> | Web | 77.1% | 81.9% | +4.8 |
> | Mobile | 83.5% | 84.9% | +1.4 |
> | Desktop | 70.4% | 71.2% | +0.8 |
>
> These preliminary results directly address the concern: stronger perception yields better downstream performance across multiple benchmarks and domains. We will include these in the revised paper.
>
> ## W3: Limited Novelty and Generalization Constraints
>
> Our core contributions center on: (1) the ScreenParse dataset, (2) the scalable Webshot pipeline, and (3) the empirical finding that dense screen-level supervision provides transferable structural priors across model families. While we acknowledge that the modeling component builds on existing architectures rather than introducing a novel one, our dataset enables apparent improvements across different datasets and baselines, demonstrating the benefits of leveraging the introduced dataset.
>
> Regarding web-centricity, we discuss this limitation in Sec. 6. We observe that despite having web-only training, models show consistent improvements on GroundCUA (PC desktop domain) across all model families tested (e.g., InternVL3 +0.178, OmniParser +0.037 PageIoU), demonstrating meaningful cross-domain transfer. We will further refine the framing in the camera-ready to be more precise about where transfer is strong versus where gaps remain.
>
> Finally, regarding detectors outperforming on localization metrics, as discussed in L355, this is expected given their detection-centric design. We see the VLM and detector approaches as complementary, and our dataset is designed to benefit both families.
>
> ## Q2: Dataset Split Hygiene
>
> Each URL is unique and rendered once. We additionally apply perceptual hashing (Hamming distance <8) to remove near-duplicate screenshots from visually similar pages, ensuring minimal content overlap across splits.
>
> ## Q3: Loss Weight Sensitivity
>
> We trained a separate run for each λ (with λ_tag = λ_loc = λ) and report results below:
>
> | λ | Val. Loss | SP PIoU | GCUA PIoU | SS-Web | SS-PC | SS-Mob |
> |---|---|---|---|---|---|---|
> | 0.5 | 0.983 | 0.556 | 0.210 | 0.390 | 0.162 | 0.044 |
> | 1.0 | 0.943 | 0.563 | 0.236 | 0.456 | 0.162 | 0.040 |
> | 1.5 | 0.921 | 0.568 | 0.236 | 0.477 | 0.153 | 0.072 |
> | **2.0†** | **0.904** | **0.565** | **0.242** | **0.493** | **0.132** | **0.040** |
> | 3.0 | **0.875** | **0.571** | 0.238 | **0.567** | **0.183** | **0.056** |
> | 4.0 | 1.407 | 0.462 | 0.233 | 0.039 | 0.018 | 0.028 |
>
> Performance improves gradually as we upweight the structure-aware tokens from λ=0.5 to 3.0, then collapses at λ=4.0, where training becomes unstable. The paper's choice, λ=2.0 (our setting, marked †), lies in this stable high-performing regime: it achieves the best GroundCUA transfer (0.242) and the second-best validation loss (0.904), supporting it as a reasonable and non-fragile choice. We will include this analysis in the revised paper.

---

> > ### Author Rebuttal · Reviewer_t63k · 2026-04-03
> >
> > Thank you for the detailed rebuttal. I appreciate the additional clarifications and analyses. The added annotation-quality audit and the new controlled downstream results are helpful, and they address my main questions well. In particular, they strengthen my confidence that ScreenParse is a useful and carefully constructed dataset, and they provide meaningful evidence that improvements in screen parsing can transfer to other settings.
> >
> > The dataset contribution is the main strength of the paper: the scale, dense supervision, and engineering effort are all valuable, and I expect the resource to be useful for the community. While, the methodological and conceptual novelty as moderate. In particular, while the paper demonstrates that the dataset can be constructed at scale and can benefit multiple model families, the deeper insight gained from the data construction and experimental analysis remains somewhat limited. The current study provides strong evidence of utility, but comparatively less understanding of what kinds of structural priors are being learned, why they transfer, or what new methodological lessons about screen parsing emerge beyond the value of dense supervision itself. However, this is consistent with my original evaluation and does not reflect an unresolved rebuttal concern.
> >
> > Adittionally, stronger evidence on downstream agent tasks would make the paper substantially more solid. The new controlled experiments are a helpful step in that direction, but a more direct demonstration of impact on agent performance would strengthen the significance claim further.
> >
> > Overall, the rebuttal resolves my main concerns. My overall assessment remains unchanged.

---

> > > ### Author Response · Authors · 2026-04-05
> > >
> > > Thank you for the constructive discussion and for indicating that your main concerns were resolved. For completeness, during the discussion period we also completed one additional desktop-oriented grounding experiment on OSWorld-G under the same controlled detector-swap setup used in our rebuttal experiments:
> > >
> > > | Detector | Text Matching | Element Recognition | Layout Understanding | Fine-grained Manipulation | Overall |
> > > | --- | ---: | ---: | ---: | ---: | ---: |
> > > | ScreenParse (YOLO) | 61.69 | 50.91 | 56.13 | 29.53 | 48.82 |
> > > | OmniParser | 59.00 | 50.30 | 53.36 | 25.50 | 46.67 |
> > > | Δ | +2.68 | +0.61 | +2.77 | +4.03 | +2.16 |
> > >
> > > Since OSWorld-G was one of the benchmark directions you explicitly suggested, we wanted to record this additional result here. It is consistent with the broader picture from our rebuttal: ScreenParse also improves performance on a harder desktop-oriented grounding benchmark. We will include this result in the revised paper.

---

### Decision · Program_Chairs · 2026-04-30

**Decision:**

Accept (regular)

**Comment:**

The reviewers agree that the paper makes a strong and valuable contribution by introducing ScreenParse, a large-scale dataset with dense screen-level annotations, along with an efficient model for structured screen parsing. A key strength consistently highlighted is that the dataset provides benefits beyond the proposed model, improving performance across multiple model families and benchmarks, and is likely to be a useful resource for the community.

The initial concerns raised by reviewers—regarding dataset quality, lack of downstream evaluation, and generalization beyond the web domain—have been substantially addressed in the rebuttal. In particular, the authors provide quantitative human validation of annotation quality and additional controlled experiments demonstrating that improvements in perception transfer to downstream grounding tasks. Following these clarifications, all reviewers indicate that their main concerns have been resolved and maintain or raise their scores accordingly.

While the modeling component is relatively incremental and some limitations remain (e.g., web-centric data and reduced performance on native UI domains), these are acknowledged and do not detract from the overall contribution. Given the strong reviewer consensus after rebuttal and the clear value of the dataset and empirical findings, the paper is recommended for acceptance.